# *Periplaneta americana* (L.) Extract PAS840 Promotes Ischemic Stroke Recovery by Inhibiting Inflammasome Activation

**DOI:** 10.3390/biology14060589

**Published:** 2025-05-22

**Authors:** Xin Yang, Canhui Hong, Tangfei Guan, Chenggui Zhang, Yongshou Yang, Peiyun Xiao, Huai Xiao, Zhengchun He

**Affiliations:** 1Yunnan Provincial Key Laboratory of Entomological Biopharmaceutical R&D, College of Pharmacy, Dali University, Dali 671000, China; 17387260406@163.com (X.Y.); hongch09@163.com (C.H.); guantangfei2021026@163.com (T.G.); chenggui_zcg@163.com (C.Z.); yangyongshou@dali.edu.cn (Y.Y.); xiaopeiyun@dali.edu.cn (P.X.); xiaohuai@dali.edu.cn (H.X.); 2National-Local Joint Engineering Research Center of Entomoceutics, Dali 671000, China; 3Engineering Research Center for Development of the Periplaneta Americana Industry of Yunnan Provincial Department of Education, Dali University, Dali 671000, China

**Keywords:** *Periplaneta americana* (L.), ischemic stroke, inflammatory body, blood–brain barrier, network pharmacology, transcriptomics

## Abstract

Ischemic stroke (IS) is a high-mortality cardiovascular disease. This study analyzed *Periplaneta americana* (L.) extract (PAS840) components using LC-MS/MS and peptidomics, and predicted its IS-related therapeutic targets by network pharmacology. In BV-2 cells after OGD/R injury, multiple techniques explored PAS840’s effects on cell functions and pathways. In a tMCAO rat model, various methods evaluated its impact on the blood–brain barrier, inflammation, and neural function. Results show PAS840 can act on multiple IS-related pathways. It inhibits the NLRP3 inflammasome pathway in BV-2 cells, reduces oxidative stress, inflammation, and pyroptosis, and improves cell viability. In tMCAO rats, it can reduce the NLRP3 inflammasome pathway, activate growth factor pathways, and protect the blood–brain barrier. These findings suggest that PAS840 is a promising drug worthy of development for the treatment of IS.

## 1. Introduction

As a major contributor to global morbidity and mortality, stroke manifests in two primary forms: ischemic and hemorrhagic subtypes [1], with ischemic stroke (IS) constituting approximately 70% of clinical presentations [2]. The multifaceted pathophysiology of IS involves intricate interactions between neuroinflammatory cascades and oxidative damage mechanisms. These pathological processes ultimately compromise blood–brain barrier integrity and induce cerebral edema formation [3].

Modern epidemiological studies have found that the incidence of IS is significantly higher in males than in females. Relevant research indicates that estrogens (e.g., estradiol) play a crucial role in the treatment of IS [4,5]. Modern medical research has found that a healthy lifestyle can effectively reduce the risk of stroke [6], and that acute anger or emotional instability is strongly associated with the occurrence of IS and cerebral hemorrhage [7]. Currently, endovascular thrombectomy (EVT) alone or in combination with intravenous thrombolysis remains the most effective treatment for IS, with no approved alternative therapy showing greater clinical effectiveness [8]. IS treatment faces a narrow time window, and patients experience more severe reperfusion injury and some neurological function damage after the operation [9]. Studies have shown that IS begins to occur when a thrombotic occlusion of the middle cerebral artery (MCA) is induced by cardiovascular or cerebrovascular diseases, or when atherosclerotic plaques lead to arterial stenosis. This pathological process triggers severe cerebral inflammation accompanied by intense oxidative stress and blood–brain barrier (BBB) damage [10]. When IS induces a brain inflammatory response, NF-κB regulates the downstream NLRP3 inflammasome and triggers the Caspase-1 cascade, activating IL-1β and IL-18 and ultimately leading to the pyroptosis of brain cells [11]. Thus, for populations at a high risk of the disease, developing drugs that can mitigate sudden disease-related injuries, alleviate brain inflammatory responses and oxidative stress damage caused by IS, and exert therapeutic effects on IS is of utmost importance [12].

*Periplaneta americana* (L.) (PA) is a cockroach species in the family Blattidae. It has been used as a traditional Chinese medicinal insect since the Shennong Bencao Jing (written between 25–220 AD) [13]. Historically employed in ethnopharmacological practices for addressing traumatic injuries, gastrointestinal disorders, and inflammatory pathologies [14], *Periplaneta americana* (L.) has been substantiated by contemporary pharmacological investigations to exhibit potent anti-pathogenic, redox-modulating, and regenerative properties in its extract (PAE) [15,16]. The preparation of medicine made from PA as raw material, “Kangfuxin Liquid”, has a history of more than 20 years and is still widely circulated and used in the market [17]. It is rich in amino acids, peptides, nucleosides, polysaccharides, and fatty acids and has significant therapeutic effects on skin burns and scalds, fatty liver, colitis, and gastric bleeding [18,19,20,21,22]. Recent studies have shown that PAE can activate the ERK/CREB/BDNF signaling pathway and exert neuroprotective and neurofunctional recovery effects on stroke [23]. Meanwhile, the commercially available drug ‘Xinmailong’ injection, which is manufactured using PA as a raw material, has been shown to effectively reduce acute myocardial infarction injury in rats [24] and act on various cardiovascular diseases [24]. Thus, based on PA’s multiple excellent pharmacological activities, we have initiated the exploration of PA’s therapeutic effects on cerebral neural diseases.

Network pharmacology, a recently emerging method for drug prediction, can analyze and assess the relationship between multicomponent drugs and diseases from multiple perspectives and levels more systematically. The theoretical study of drug effects is consistent with the holistic concept of traditional Chinese medicine, and its advent has provided more scientific and reliable methods for exploring the mechanisms of complex Chinese herbal medicine systems in the treatment of diseases [25].

Based on the results of the previous research conducted by our research group, this study reveals that PAS840 (an extract of PA, named according to the characteristics of the extraction process in Section 2.1) can effectively alleviate the inflammatory response and oxidative stress damage induced by glutamate in PC12 nerve cells [26]. It can also significantly reduce the area of cerebral infarction in rats and the neuronal apoptosis of brain tissue, and regulate the ecological balance of gut microbiota in rats [27]. Therefore, we further analyzed the components of PAS840 using liquid chromatography-tandem mass spectrometry (LC-MS/MS) technology and peptidomics, and predicted the disease targets of the identified components using network pharmacology approaches. Oxygen–glucose deprivation/reoxygenation (OGD/R) and transient middle cerebral artery occlusion (tMCAO) are widely-used contemporary models for IS [28,29]. Through in vitro and in vivo experiments, they can effectively mimic the process of oxygen and glucose deprivation in nerve cells and brain tissue following IS occurrence in animals or humans. As an internationally recognized in vitro experimental sample, microglial cells (BV-2) exhibit high consistency with primary cells in core phenotypes such as neuroinflammatory responses and phagocytic function [30]. In this study, the OGD/R model was constructed using BV-2 cells, and the tMCAO model was established using rats to verify the effects of PAS840 on oxidative stress, the NLRP3/Caspase-1/IL-1β pyroptosis pathway, BBB integrity, and neural function repair. This study provides a scientific basis for the potential role of PAS840 in the prevention and treatment of IS.

## 2. Materials and Methods

### 2.1. Naming and Extraction Process of PAS840

PAS840: The dried PA body powder of the PA was extracted with 90% ethanol at a ratio of five times the weight of the PA material, and the ethanol solution was concentrated under reduced pressure to obtain an ethanol extract. The ethanol extract was then dried to obtain a dehydrated extract. The defatted extract was adsorbed on an S-8 macroporous adsorption resin column, and the resin column was first washed with water and then washed with 40% ethanol to elute the resin column. The ethanol eluate was collected, concentrated at 60 °C under reduced pressure, and lyophilized to obtain the PAS840 sample.

### 2.2. Materials

Materials included Cell Counting Kit-8 (CCK-8) (Aladdin, Wuhan, China, Lot No.: L2-128-203), 96-well/24-well/6-well cell culture plates (NEST, Gretna, LA, USA, Lot No.: 052024BL01), MCC950 NLRP3 inhibitor (MedChemExpress, Shanghai, China, Lot No.: CP-456773), BV-2 mouse microglia specific culture medium/sugar-free DMEM culture medium (Wuhan PNase Life Science Technology Co., Ltd., Wuhan, China, Lot No.: WH4424E112/WHAA24P244), PBS buffer/4% paraformaldehyde/hematoxylin (Wuhan Saive Biotechnology Co., Ltd., Wuhan, China, Lot No.: GP21090061782/GP2-306304/G1004), crystal violet dye (Baiyuntian, Shaoguan, China, Lot No.: C0121-100), Transwell (Costar, Arlington, VA, USA, Lot No.: 02422025), Evans Blue Dye (MACKLIN, Shanghai, China, Lot No.: C14442662), triphenyltetrazolium chloride (TTC) stain (Aladdin, Shanghai, China, Lot No.: L2128203), qPCR detection kit (Aicrui Biotechnology Co., Ltd., Qingdao, China, Lot No.: a6a0298), primers (Shenggong Bioengineering Co., Ltd., Shanghai, China, Lot No.: 300371063), malondialdehyde (MDA) detection kit/superoxide dismutase (SOD) detection kit (Boxbio, Beijing, China, Lot No.: 10124071220/10124070650), total nitric oxide synthase (NOS) subtype detection kit/reactive oxygen species (ROS) detection kit (Nanjing Jiancheng Bioengineering Institute, Nanjing, China, Lot No.: 2024107/2023128), tMCAO monofilament/bupivacaine hydrochloride (Beijing Xichong Biotechnology Co., Ltd., Beijing, China, Lot No.: 20230506), neutral Gum/HE stain (Wuhan Bolf Biotechnology Co., Ltd., Wuhan, China, Lot No.: B0044/BHO001), NLRP3/NF-κB antibody/secondary antibody (Wuhan Punose Life Science Technology Co., Ltd., Wuhan, China, Lot No.: GB114320/GB11142/GB21303), and isoflurane (Shandong Ante Herding Technology Co., Ltd., Jinan, China, Lot No.: 2023090502).

### 2.3. Network Pharmacology Analysis of IS by PAS840

#### 2.3.1. Identification of the Composition of PAS840 Substances

The determination of small and medium molecular compounds in PAS840 samples: Sample treatment: Mix an appropriate amount of sample with 600 µL of 2-chloro-L-phenylalanine (4 ppm) methanol solution, swirl for 30 s, grind at 50 Hz for 120 s using a tissue grinder (MB-96, wall), ultrasound at room temperature for 10 min, centrifuge at 412,000 rpm for 10 min at 4 °C, and filter supernatant through a 0.22 μm membrane. Analysis: Use a Thermo Vanquish (Thermo Fisher Scientific, Waltham, MA, USA) ultra-high performance liquid phase system with ACQUITY UPLC^®^ HSS T3 (2.1 × 100 mm, 1.8 µm) column (Waters, Milford, MA, USA) with a flow rate of 0.3 mL/min, column temperature of 40 °C, and sample size of 2 μL. Mobile phase: In positive ion mode, 0.1% acetonitrile formate (B2) and 0.1% water formate (A2); in negative ion mode, acetonitrile (B3) and 5 mM ammonium formate water (A3). Gradient elution is shown in Table 1. Mass spectrometry: Use Thermo Q Exactive (Thermo Fisher Scientific, Waltham, MA, USA), ESI, and positive and negative ion modes with a positive ion spray voltage of 3.50 kV, negative ion spray voltage of -2.50 kV, sheath gas of 40 arb, auxiliary gas of 10 arb, and capillary temperature 325 °C. Conduct the first-level full scan at 70,000 resolution, m/z 100–1000 range. Then, perform second-level cracking by HCD, with a collision energy of 30 eV, and second-level resolution of 17,500, and break the first 10 ions before signal collection. Use dynamic exclusion to remove unnecessary MS/MS information.

PAS840 peptide component detection: Mobile phase: Liquids A (100% water and 0.1% formic acid) and B (80% acetonitrile and 0.1% formic acid). Sample prep: Dissolve lyophilized powder in 10 mL of A. For liquid–liquid detection, inject 1 µg supernatant after 20 min at 4 °C and 14,000 rmp centrifugation. Chromatography: Elution is shown in Table 2. Mass spec: Use Nanospray FlexTM (NSI) ion source and Q Exactive HF-X (Thermo Fisher Scientific, Waltham, MA, USA). Use an ion spray voltage of 2.4 kV, and ion transfer tube temp of 275 °C. Acquire in data-dependent mode. 1st stage MS: full scan m/z 100–1500, resolution 120,000 (200 m/z), AGC 3 × 10^6^, and max C-trap inj time 80 ms. Fragment the parent ion (highest intensity) by HCD. 2nd MS: resolution 15,000 (200 m/z), AGC 5 × 10^4^, max inj time 45 ms, and peptide frag/collision energy 27%. Detection: Use raw data (.raw) for MS. Retrieve peptide sequences by de novo analysis with Peptides and De and Novo-only Peptide databases.

#### 2.3.2. Acquisition of Disease Targets Corresponding to PAS840 Components

To obtain the related protein information, the peptide sequences that scored highest on the Peptides database analysis (−10 lgP value) and those that scored higher on de novo-only peptide analysis (>95 confidence level) were chosen and added to the Emboss database (https://www.ebi.ac.uk/Tools/seqstats/emboss_pepstats/, accessed on 8 January 2024). Peptide sequences with a charge value of ≥0 and an isoelectric point value of ≤12 were screened according to the conditional characteristics of food-borne anti-inflammatory peptides and neuroprotective peptides [31,32]. The peptide sequence numbers were then entered into the NoverPor (https://www.novoprolabs.com/, accessed on 12 January 2024) web tool to convert the sequence numbers to the smile format, and the similarity ensemble approach (SEA) (https://sea.bkslab.org/, accessed on 12 January 2024) was used to obtain the target protein for each peptide prediction. After eliminating duplicates, the target protein information of the peptides and small molecules was combined to obtain the desired information.

#### 2.3.3. IS Disease Target Acquisition

To identify the intersection of PAS840 targets with IS, the search term “cerebral ischemia” was chosen, and the OMIM (https://omim.org/, accessed on 12 January 2024), GeneCards (http://www.genecards.org/, accessed on 12 January 2024), and DisGeNET (https://www.disgenet.org/search, accessed on 12 January 2024) databases’ search results were combined. Duplicate disease targets were removed, and the disease targets and PAS840 targets were imported into Venn 2.1.0.

#### 2.3.4. Disease Target Interaction Network Construction

The intersection illness target was entered into the STRING Protein Interaction Retrieval Prediction Database (https://string-db.org/, accessed on 12 January 2024). The relevant data were produced by hiding the disconnected nodes in the network and setting the remaining parameters to their default values. The protein network was analyzed using Centiscape 2.2 of the Cytoscape 3.9.1 software. The core protein target was screened using default screening conditions of degree (>39.764434), closeness (>0.001091), and betweenness (>498.244804), and a protein–protein interaction (PPI) network map was created.

#### 2.3.5. Gene Ontology and KEGG Enrichment Analysis

The DAVID database (https://david.ncifcrf.gov/, accessed on 12 January 2024) was used to import disease target information, and the OFFICIAL_GENE_SYMBOL was chosen in DAVID for the Kyoto Encyclopedia of Genes and Genomes (KEGG) pathway enrichment and biological process gene ontology (GO) analyses. The online bioinformatics platform Microbiosense (http://www.bioinformatics.com.cn/, accessed on 12 January 2024) was used to visualize the enrichment outcomes.

#### 2.3.6. Component–Pharmacological Target–Disease Pathway Network Construction

The PAS840 target, intersection target, and KEGG pathway data were acquired and utilized to create a composition–pharmacodynamic target–disease pathway network worksheet. The data were then integrated into a tape file and imported into Cytoscape 3.9.1 software to create a network diagram of the composition–pharmacodynamic target–disease pathway. Simultaneously, node degree values were computed using Centiscape 2.2, a Cytoscape 3.9.1 plug-in, to assess the connections between the sample components, pharmacodynamic targets, and disease pathways.

#### 2.3.7. Molecular Docking

Target proteins with the highest degree of interaction and matching active components that were filtered out of the PPI network were subjected to molecular docking. Structural files for the target proteins and active ingredients were obtained from the PDB database (https://www.rcsb.org/, accessed on 23 January 2024). PyMoL 4.6.0 software was used to dewater and deligand the proteins, and the active sites were chosen for molecular docking by focusing on the original ligands of the target proteins. Molecular docking was performed using the AutoDock Vina 1.5.7 software to confirm the binding activity between the key target proteins and matching active components, with the search space volume >27000 Angstrom3 (using random seed files and each protein box size, see raw data), record affinity value. PyMoL 4.6.0 software was used to visualize the outcomes.

### 2.4. In Vitro Experiments

#### 2.4.1. The Effect of PAS840 on the Survival Rate of BV-2 Cells Was Detected by cck-8

Mouse microglial cells (BV-2) were provided by Wuhan Punose Life Science Technology Co. Ltd. (Lot number: 8R58KVUP17, Wuhan, China), the BV-2 cells used in the experiments were maintained within 10 passages. The logarithmic growth phase of BV-2 cells was adjusted to a cell suspension with a density of 1.5 × 10^5^ cells/mL. A total of 100 μL of the cell suspension was seeded into 96-well plates and incubated at 37 °C and 5% CO_2_ for 24 h. Then, 100 μL of PAS840 complete culture medium solution at concentrations of 20, 40, 80, 160, 320, 640, and 1280 μg/mL were added to the corresponding wells (the drug is completely soluble in the culture medium without interference from phosphate-buffered saline or dimethyl sulfoxide), and the Control group was given 100 μL of the complete culture medium. After culturing for 24 h, 10 μL of CCK-8 was added to each well. The reaction was allowed to proceed for 1 h, and the optical density (OD) at 450 nm was measured using a multiwavelength microplate reader. Cell viability (%) and cell activity were calculated, and the safe dose of PAS840 was determined.

#### 2.4.2. Experimental Grouping

Based on the CCK-8 assay results in Section 2.4.1, select the dosage range that can effectively enhance cell viability without inhibiting cell survival rate as the administration dosage for subsequent experiments. To validate the effect of PAS840 on injured cells in the OGD/R model, we set up the following groups: the Control group without treatment, the OGD/R group, the NLRP3 inhibitor group (10 nm/L, MCC950 + OGD/R) [33], and the PAS840-treated groups at low, medium, and high doses (40 μg/mL, PAS840 L + OGD/R; 80 μg/mL, PAS840 M + OGD/R; and 160 μg/mL, PAS840 H + OGD/R).

#### 2.4.3. Establishment of OGD/R Model and Detection of Survival Rate of Injured Cells

Logarithmic proliferation period BV-2 cells were obtained, the cell density was adjusted to 1.5 × 10^5^ cells/mL cell suspension, 100 μL of which was inoculated into 96-well plates, and the cells were cultured at 37 °C and 5% CO_2_ for 24 h under normal conditions. The Control group continued to use a complete culture medium under normal conditions, and the OGD/R group was replaced with glucose-free DMEM. The sugar-free DMED containing PAS840 was replaced in the PAS840 treatment group and cultured separately in a 95% N_2_, 5% CO_2_ environment for 4, 6, and 8 h before being replaced with a complete medium and further cultured for 24 h. Refer to Methods Section 2.4.1 for how the CCK-8 assay was used to detect cell viability and determine the subsequent modeling time. According to the grouping conditions described in Methods Section 2.4.2, cells were seeded in 6-well plates and observed under an inverted microscope after 12 h of drug administration and modeling.

#### 2.4.4. The Migration Ability of PAS840 to OGD/R-Injured Cells Was Detected

According to the grouping conditions specified in Section 2.4.2, the cells were cultured in 6-well plates, treated, and modeled according to the method specified in paragraph 2.4.3. In a 24-well plate, 500 μL of medicated complete medium was added according to the grouping conditions specified in Section 2.4.2, and the cells were suspended in DMEM medium without serum and adjusted to a cell density of 2.5 × 10^5^ cells/mL. A total of 200 μL of the cell suspension was then placed on the upper chamber of the Transwell, and the Transwell was placed in the 24-well plate. The cells were cultured at 37 °C and 5% CO_2_ for 48 h. After removing the Transwell plate, the cells were fixed with paraformaldehyde and stained with crystal violet for 20 min. The cells were then observed and counted under a microscope.

#### 2.4.5. Biochemical Indicators Testing

According to the grouping conditions specified in Section 2.4.2, the cells were cultured in 6-well plates, treated, and modeled according to the method specified in paragraph 2.4.3. According to the operation instructions of the MDA, SOD, NOS, and ROS kits, take 5 × 10^6^ cells from each group of samples. Add the cell lysis buffer provided in the kits to fully lyse the cells. Then, add the working solution provided in the kits as described in the instructions. After a sufficient reaction, take 200 μL from each group of samples and add them to a 96-well plate. Measure the OD values and fluorescence values using a multi-wavelength microplate reader as required by the instructions, and calculate the expression levels of MDA, SOD, NOS, and ROS. All steps were performed in accordance with the manufacturer’s instructions in the test kit manuals.

#### 2.4.6. mRNA Expression Detection and Analysis by RT-qPCR in BV-2 Cells

According to the grouping conditions specified in Section 2.4.2, the cells were seeded into 6-well plates, treated, and modeled according to the method specified in Section 2.4.3. Add 1 milliliter of Working Solution 1 from the real-time quantitative polymerase chain reaction (RT-qPCR) kit to each group of cells. Extract the total RNA as instructed, and then measure the purity and concentration of the RNA using an ultraviolet spectrophotometer. The concentrations of the sample RNA were standardized, and cDNA was synthesized from total RNA through reverse transcription. The amplified products were analyzed by RT-qPCR using β-actin as an internal reference. The relative quantification of the target gene was performed using the 2^−ΔΔCT^ method. The primer sequences are listed in Table 3.

#### 2.4.7. Transcriptomic Sequencing

Following cell inoculation into 6-well plates according to Section 2.4.2 grouping protocols, modeling and drug treatments were performed as per Methods Section 2.4.3. Total RNA from Control, OGD/R, and PAS840H + OGD/R groups was quantified and qualified (≥1 µg). RNA libraries were constructed using the NEBNext Ultra II RNA Library Prep Kit (Illumina) (New England Biolabs Inc; Ipswich, MA, USA): Fragmented mRNA templates were reverse-transcribed into double-stranded cDNA with random oligonucleotide primers. After purification and end repair, 3’-terminal “A” overhangs were added. AMPure XP beads were employed to select cDNA fragments (400–500 bp) for final library preparation. Equimolar pooled libraries were diluted to appropriate concentrations and sequenced on an Illumina platform (PE150 mode).

Raw FASTQ data were generated from image files via the sequencing system software. HISAT2 (v2.1.0) indexed the reference genome and aligned clean paired-end reads, while HTSeq (v0.9.1) quantified gene read counts. Differential expression analysis was conducted using DESeq (v1.38.3). GO enrichment analysis (via topGO) calculated *p*-values for annotated terms through the hypergeometric distribution method (significance threshold: *p* < 0.05). KEGG pathway analysis implemented with ClusterProfiler statistically evaluated pathway-level gene enrichment using the same significance testing framework.

### 2.5. In Vivo Experiments

#### 2.5.1. Grouping and Dosing of Animals

In our preliminary studies, we found that female rats exhibited significantly better neurological function than male rats after model establishment. To minimize the interference of estrogen on the therapeutic effect of the disease and better compare drug efficacy, 36 male Sprague-Dawley (SD) rats were used in the experiment. These rats were purchased from Beijing Sipfer Biotechnology Co., Ltd. (License No.: SCXK 2019-0010). The housing conditions were SPF grade, selected rats were 15 weeks of age, and the weight of the rats at the time of modeling was 250 ± 20 g. All rats were placed in a strictly controlled environment (dark/light cycle: 12 h; temperature: 24 ± 2 °C; relative humidity: 65% ± 5%), and each rat was given 60Co irradiated sterilized rat food and purified water daily. The animals involved in this study were approved by the Experimental Animal Ethics Committee of the Dali University (approval number: 2024-PZ-002).

The rats were randomly divided into four groups, with 9 rats per group. They were named the Sham group, Model group, PAS840 L group (gavage: 30 mg/kg), and PAS840 H group (gavage: 120 mg/kg), The gavage dosage was selected based on previously published papers by our research team [27]. The drugs were dissolved in normal saline for administration, with equivalent volumes of saline administered to both the Sham and Model groups. To investigate the effects of drugs on preventive and adjuvant treatment in high-risk populations for diseases, after 7 days of adaptive feeding, the tMCAO model was established in rats 12 h after the first drug administration. Following model establishment, the rats received the drug for an additional 7 days, with administrations conducted at fixed time points. The Sham group was administered an equivalent volume of normal saline. Based on previous studies by our team, PAS840 reached its absorption peak in rat cerebrospinal fluid 6 h after administration. Therefore, all rats were sacrificed 6 h after the last drug administration, and tissue samples were collected from the rats.

#### 2.5.2. tMCAO Model Establishment

To ensure experimental uniformity, all model establishments for samples were independently completed by the first author within 24 h. A rat tMCAO model was established by a suture occlusion of the right middle cerebral artery. The right common carotid artery (CCA), external carotid artery (ECA), and internal carotid artery (ICA) were exposed via a midline neck incision. Anesthesia was induced with 4% isoflurane and maintained by continuous inhalation at 1.75%. Post-anesthesia, the ECA was transected, and the tMCAO monofilament was inserted through the ECA to a depth of (18 ± 1) mm within the ICA. After wound suturing, the tMCAO monofilament was removed at 90 min. In the Sham group, only skin incision and vessel dissection were performed before suturing. Postoperatively, bupivacaine hydrochloride was administered to all rats for analgesia. The operation duration for a single rat was 20 to 25 min, and all sample sizes were selected to meet the animal welfare requirements specified by the Ethics Committee of Dali University and the minimum statistically significant sample size.

#### 2.5.3. Weight Monitoring

On the day of modeling, as well as after the model was completed, the fasting body weight of each group of rats was monitored daily, and a dynamic weight monitoring curve graph was drawn.

#### 2.5.4. Neurological Function Measurement

Twenty-four hours post-modeling, neurological function in each rat group (*n* = 6) was assessed via the Zea Longa scale [34]. The scoring criteria was as follows: 4 (complete paralysis), 3 (unilateral paralysis), 2 (paralyzed-side dragging during walking), 1 (torso twisting with impaired forelimb extension upon tail lift), and 0 (no deficits). Scores ≥1 confirmed successful nerve injury modeling.

#### 2.5.5. Evans Blue Dyeing

Each group of rats (*n* = 3) was injected with 2% Evans blue (EB) solution via the tail vein. One hour later, they were anesthetized with isoflurane and perfused with saline. After perfusion was completed, the entire rat brain was removed and photographed. Then, 100 mg of brain tissue was ground using 1 mL of formamide solution, incubated at 37 °C for 24 h, and the OD value at 620 nm was measured using a multi-wavelength microplate reader. A standard curve was plotted using the EB standard solution, and the EB content of all samples was calculated.

#### 2.5.6. TTC Staining

The brain tissue samples from rats (*n* = 3) were isolated and subjected to TTC histochemical processing. Following cryopreservation at −20 °C for twenty minutes, intact brains were sectioned coronally into five sequential 2 mm thick slices. These tissue sections were subsequently incubated with 4% TTC staining solution under controlled conditions (37 °C, avoiding light) for 20 min.

#### 2.5.7. HE Staining of Brain Tissue

For each group, the brains were collected (*n* = 3). After wax-embedding, sectioning, dewaxing, and hydration, the brain sections were placed on glass slides. Then, they were stained with hematoxylin and eosin (HE) staining solution, dehydrated, cleared, and sealed. Subsequently, the pathological morphology of the cerebral cortex was observed under a microscope.

#### 2.5.8. mRNA Expression Detection and Analysis by RT-qPCR in Brain Tissue

Taking the cortical part of the infarcted brain tissue of rats in Section 2.5.7 (*n* = 3), and following the experimental method described in Section 2.4.5, β-actin was used as an internal reference to perform a relative quantitative analysis of the target gene using the 2^−ΔΔCT^ method. The primer sequences are listed in Table 4.

#### 2.5.9. Immunohistochemical Analysis of NLRP3 and NF-κB Expression

Take Section 2.5.7 rat brain tissue samples (*n* = 3), section them with paraffin, dewax them to water, perform antigen repair, block endogenous peroxidases, incubate with 3% rabbit serum at room temperature for 30 min, incubate with the primary antibody at 4 °C overnight, incubate with the secondary antibody at room temperature for 50 min, add a freshly prepared DAB chromogen solution, counterstain with hematoxylin for 3 min, dehydrate, mount the sections, and observe and photograph them under a microscope. Use ImageJ 1.53t for statistical analysis.

### 2.6. Statistical Analysis

All data were analyzed using the ParsonalBio GenesCloud platform (https://www.genescloud.cn/cloudClassroom, accessed on 18 November 2024) and SPSS 27. For statistical analysis in SPSS 27, both “Tukey” and “Dunnett-t” were selected as post-hoc multiple comparison methods in the ANOVA results window. Analysis results assumed homogeneity of variance with a significance level set at 0.05, where *p* < 0.05 was considered statistically significant. When homogeneity of variance was confirmed, the Tukey method was used for analysis; when homogeneity of variance was not confirmed, the Dunnett-t method was applied instead. The analysis results were visualized and plotted using GraphPad Prism 8.0.2 and the Parsonalbio Genescloud platform. Cytoscape 3.9.1 was used to construct relevant network diagrams.

## 3. Results

### 3.1. Network Pharmacological Analysis of PAS840 on IS

#### 3.1.1. PAS840 Component Analysis Results

LC-MS/MS analysis showed that PAS840 contained 223 small-molecule compounds (Figure 1) and 16,788 peptide sequences. Notably, 2035 and 14,753 of these sequences are contained in the Peptides and De Novo-only Peptides databases, respectively. Based on the set of peptide screening criteria, 50 peptide sequences were screened from 106 peptides with a top −10 lgP value from the Peptides database, and 128 peptide sequences were screened from 259 peptides with >95% confidence from De Novo-only Peptides. A total of 178 polypeptide sequences were identified.

In the analysis of small-molecule compounds, 46 categories were found, and carboxylic acids and their derivatives accounted for 15.7% of all small-molecule compounds. This was followed by benzene and its substituted derivatives (14.34%), fatty acids (7.17%), and phenols (5.83%). The remaining 41 categories small-molecule compounds and detailed molecular information can be found in the Appendix A.

#### 3.1.2. Core Target Acquisition

A total of 999 therapeutic targets were identified by predicting 223 small-molecule compounds in PAS840 and 178 polypeptide sequences. By merging data from the disease database, 4414 disease targets were acquired, and 435 intersection proteins were obtained using a Venn diagram (Figure 2A). A protein–protein interaction (PPI) network analysis was performed on the obtained intersection targets, and 93 core protein nodes were screened for protein interaction networks with 1944 edges (Figure 2B). Among them, MMP9, STAT3, Caspase3, MAPK3, CD4, NOS3, IL-1B, and IL-6 (core proteins closely related to inflammation), and Bcl-2, Akt, EGFR, and HIF-1A (core proteins with positive effects on IS treatment) strongly correlated with IS.

#### 3.1.3. Enrichment Analysis

After subjecting the intersection proteins to GO (Figure 3A) and KEGG (Figure 3B) enrichment analyses, the biological process (BP) information from the GO enrichment analysis indicated that the components of PAS840 are involved in positively regulating the cytoplasmic calcium ion concentration, MAPK cascade, phospholipase C-activating G-protein coupled receptor signaling pathway, and resistance to exogenous stimuli. Moreover, PAS840 had a positive effect on therapeutic targets such as cytosolic calcium ion concentration, inflammatory response, and anti-stress manifestation in IS.

In the cellular component (CC) and molecular function (MF) information, PAS840 primarily acted on intercellular messaging, participated in apoptosis, and produced effective cytoprotective and anti-stimulatory effects at the onset of IS.

Finally, the KEGG enrichment analysis of its pathway revealed that PAS840 can regulate the TNF signaling pathway, neuroactive ligand–receptor interaction pathway, Th17 cell differentiation, and other pathways closely related to IS, and can exhibit anti-inflammatory, anti-apoptotic, and neuroprotective properties, and positively regulate the calcium signaling pathway, among other effects.

#### 3.1.4. Drug–Target–Pathway Correlation Network Analysis

The collected KEGG target information, cross-protein information, and peptide data were presented through the network and sorted according to the degree of relevance (Figure 4). The five interregional links were closely correlated, indicating that PAS840 was highly correlated with the therapeutic target tubes of IS. Among them, the small molecular components such as PA72, PA30, PA37, and PA4, and the peptides such as peptide173, peptide108, peptide47, and peptide114 in the network diagram have close associations with the core proteins, such as STAT3, ACE, Caspase3, MAPK3, CD4, NOS3, IL-1B, and IL-6, that are important in relation to IS, as well as with the signaling pathways, such as hsa04080: neuroactive ligand–receptor interaction, hsa04020: calcium signaling pathway, and hsa04210: apoptosis, that are also important in relation to IS in KEGG.

#### 3.1.5. PAS840 Has Good Docking Activity with Multiple Disease Targets of IS

Based on the binding energy of the molecules after docking, the lower the binding energy of the ligand to the receptor, the more stable the binding and the stronger the binding activity. Twelve target proteins with high degree values and high IS correlations obtained in the PPI analysis were selected for docking with 20 small molecules (Table 5). Another 10 target proteins were selected for molecular docking with peptide sequences with the top 10 deg values (Table 6) under the same conditions, and the docking results showed that 93.33% of the small molecules had binding energies < −5 kcal/mol to their targets, and 29.17% of the small molecules had binding energies < −7.0 kcal/mol to their targets; 100% of the peptide sequences in the peptides had binding energies < −5 kcal/mol to their targets, and 76.00% of the peptide sequences had binding energies < −7.0 kcal/mol to their targets. The docking results displayed in heat maps (Figure 5A,C) show that the active components of PAS840 bind strongly to the IS targets. PyMoL software was used to visualize and present the small-molecule compounds and peptides with binding energies < −9 kcal/mol (Figure 5B,D).

### 3.2. Results of In Vitro Experiments

#### 3.2.1. Effect of PAS840 on the Survival Rate of BV-2 Cells

By comparing the effects of PAS840 at different concentrations on BV-2 cells after incubation for 24 h, it was found that PAS840 concentrations of 20, 40, 80, and 160 μg/mL could effectively promote BV-2 cell proliferation (*p* < 0.001), while the concentration of 640 μg/mL began to inhibit BV-2 cell proliferation and produce cytotoxicity (Figure 6A). Therefore, 40, 80, and 160 μg/mL of PAS840 were selected as the subsequent doses.

#### 3.2.2. OGD/R Injury Model

By setting three different time points for OGD-induced injury, it was found that the longer the OGD duration, the lower the viability of BV-2 cells compared to the Control group. However, after OGD/R injury, the PAS840 treatment group significantly improved the cell viability of BV-2 cells in all three time points (Figure 6B). By comparing the morphology of cells in the six groups, it was found that OGD/R significantly decreased cell density and induced obvious cell differentiation. However, PAS840 treatment significantly increased cell density, increased the number of high-activity cell clusters, and reduced cell differentiation (Figure 6C). Based on the cell survival rate of the OGD/R group in the experiment, the 7 h OGD time with a survival rate closer to 50–60% was selected as the OGD injury time for the subsequent experiment.

#### 3.2.3. Effect of PAS840 on Cell Migration

After OGD/R injury, the migration ability of BV-2 cells was significantly inhibited in the OGD/R group compared to that in the Control group, whereas the migration ability of BV-2 cells was significantly improved after treatment with MCC950 and PAS840 (*p* < 0.05) (Figure 7). This indicates that the administration of PAS840 can effectively improve the cell migration ability, and the effect increases with an increase in the drug dose, suggesting that PAS840 may play a positive role in restoring the function of BV-2 cells after OGD/R injury.

#### 3.2.4. Effects of PAS840 on MDA, SOD, NOS, i NOS, and ROS Levels in Injured Cells

After OGD/R injury, the levels of MDA, as well as T NOS, and i NOS in OGD/R-injured cells were significantly increased compared to those in the Control group, while the fluorescence intensity of ROS was elevated. SOD activity was significantly lower. However, after treatment with MCC950 or PAS840, the activity of SOD significantly increased (*p* < 0.01), whereas the levels of MDA, NOS, and ROS significantly decreased (*p* < 0.01) (Figure 8). This indicated that PAS840 could effectively inhibit the oxidative stress response caused by OGD/R injury in cells.

#### 3.2.5. Effect of PAS840 on mRNA Expression of Mmp9, Nf-kb1, Nlrp3, Caspase1, Il-1β, Jak3, and Mapk3 in Injured Cells

After OGD/R injury, according to the results of RT-qPCR (Figure 9), compared with the Control group, MCC950 or PAS840 intervention could significantly reduce the mRNA expression of *Mmp9*, *Nf-kb1*, *Nlrp3*, *Caspase1*, *Il-1β*, *Jak3*, and *Mapk3* in OGD/R-injured cells (*p* < 0.05), indicating that PAS840 can effectively inhibit the expression of relevant factors in the NLRP3/Caspase1/IL-1β inflammasome pathway caused by OGD/R injury.

#### 3.2.6. Transcriptomic Analysis of PAS840 on Injured Cells

##### Differential Analysis of RNA Expression in OGD/R-Injured Cells by PAS840

BV-2 cells from the Control, OGD/R, and PAS840H + OGD/R groups were collected for RNA detection and the identification of degradation. PCA and a correlation analysis heat map (Figure 10A,B) showed that each group had a strong correlation within itself and a large gap between the OGD/R group and the other two groups, indicating that the experimental model had good stability. The cluster heat map (Figure 10C) was further drawn for analysis, and the results showed that compared with the OGD/R group, the gene expression in the PAS840H-treated group was closer to that in the Control group to a certain extent.

Further differential gene analysis (Figure 10D) revealed that the Control group versus the OGD/R group had 1879 differential genes, with 561 upregulated and 1318 downregulated genes. Meanwhile, the PAS840 group, compared to the OGD/R group, had 879 significantly differentially expressed genes, with 119 upregulated and 760 downregulated genes. The volcano plots are shown in Figure 10E,F.

##### Differential Gene Analysis of OGD/R-Injured Cells by PAS840

Venn and UpSet diagrams (Figure 11A,B) were created for the analysis of differential gene expression in each group. A total of 72 identical differential genes were identified across the three groups, and the differential gene expression between the PAS840 treatment group and the OGD/R group was closer to that between the Control group and the OGD/R group. The PAS840 treatment group versus the OGD/R group exhibited 94 unique differential genes, which may represent the key genes expressed following PAS840’s action on OGD/R. A genomic circle map visualized the expression of all differential genes (Figure 11C).

##### Protein Interaction Analysis of Differential Genes

Differences in gene expression between the Control group and the OGD/R group and the PAS840 treatment group and the OGD/R group were analyzed using a protein interaction network map (Figure 12A,B). The results showed that the expression of genes closely related to inflammation, such as *Tnf*, *Nos*, *Cxcl2*, *Mapk3*, *Mapt*, *Jak3*, *Cd4*, and *Cd46*, was downregulated in both comparison groups. These genes exhibited a high degree of consistency with the core targets identified by network pharmacology, and the unique differential genes between the PAS840 treatment and OGD/R groups were analyzed. The PAS840 treatment group showed significantly downregulated *H2ac11*, *H2bc8*, *H4c11*, *H3c4*, *H2ac6*, *H3c6*, *Dusp1*, *Selp*, *Rras*, *H2AC11*, *Serpine1* and *Junb*; *Fosl1* and *Cdh5* were upregulated. These changes in gene expression suggested that PAS840 administration had a positive regulatory effect on IS.

##### Transcriptomic Enrichment Analysis

GO (Figure 12C) and KEGG enrichment analyses (Figure 12D) of the PAS840 treatment group versus the OGD/R group. The GO enrichment analysis results showed that PAS840 administration had a positive impact on OGD/R-injured cells in terms of their response to hypoxia, resistance to chemical and non-biological stimuli, cellular signal transduction, positive regulation of cell development processes, and protein binding. KEGG enrichment analysis indicated that PAS840 administration had a positive impact on the HIF-1 signaling pathway, chemokine signaling pathway, MAPK signaling pathway, Rap1 signaling pathway, and calcium signaling pathway in OGD/R-injured cells. Combined GO and KEGG analysis results suggested that PAS840 can produce positive and active effects on enhancing cell viability, resisting external stimuli, positively regulating cytokines, controlling inflammation, and controlling calcium overload, which is highly consistent with the network pharmacology enrichment analysis results.

##### Transcription Factor Analysis

Transcription factors (TF) and differential genes predicted for TF were counted. The zf-C2HC, Homeobox, BHLH, p53, and TF-bZIP transcription factor families exhibited a high number of expressed genes in the three sample groups (Figure 12E). After comparing the differences in TF expression between the PAS840 treatment group and the OGD/R group, it was found that PAS840 had a significant association with the expression of the zf-C2HC transcription factor family (Figure 12F), and that zf-C2H2 plays an important role in regulating cell apoptosis and participating in the treatment of neurological diseases. Therefore, combined with the transcriptomic data analysis, it was indicated that the administration of PAS840 positively inhibits the inflammation and apoptosis of OGD/R-injured cells, enhances cell activity, and regulates cytokines.

### 3.3. Results of In Vivo Experiments

#### 3.3.1. Effect of PAS840 on Body Weight of IS Rats

Through dynamic body weight measurements (Figure 13A), it was found that after the tMCAO model was established, the weight of the rats showed a marked decrease, but PAS840 administration reversed this phenomenon more quickly, especially in the PAS840 H group, where the weight began to increase significantly on the third day after modeling, whereas the weight in the Model group did not begin to increase until the fifth day after modeling. This indicates that administering PAS840 can effectively alleviate weakened appetite and digestive ability caused by IS and accelerate the recovery of physical strength after IS.

#### 3.3.2. The tMCAO Model Was Successfully Established

The Zea Longa scoring results from each group of rats demonstrated that all rats incorporated into the tMCAO model exhibited neurological function scores ≥ 1, indicating the successful establishment of the tMCAO model (Figure 13B).

#### 3.3.3. Effect of PAS840 on the Function of the BBB in IS Rats

Through the Evans blue (EB) staining experiment (Figure 13B), it was observed that after transient middle cerebral artery occlusion (tMCAO) modeling, the function of the BBB in rats was severely compromised, and the content of EB in the brain tissue significantly increased. However, the administration of PAS840 could effectively mitigate the injury to the BBB and decrease the content of EB in the brain tissue (*p* < 0.05), indicating that it has a certain protective and restorative effect on the function of the BBB.

#### 3.3.4. PAS840 Reduces the Area of Infarction

According to the results of TTC staining, PAS840 can effectively reduce the area of cerebral infarction. Due to limitations in staining techniques, the staining results are shown in Appendix A.

#### 3.3.5. PAS840 Can Reduce Brain Tissue Damage

The results of HE staining of brain tissue sections showed that when IS occurred, it led to severe tissue vacuolation and inflammatory cell infiltration in the cerebral cortex. After the administration of PAS840, this condition was improved, and the tissue state became more similar to that of the Sham group (Figure 13C). This indicates that the administration of PAS840 can effectively alleviate brain tissue damage caused by IS.

#### 3.3.6. Effects of PAS840 on Nf-kb1, Nlrp3, Caspase1, Il-1β, Vgf, Bdnf, Erbb4, and Nrg1 in the Brain Tissue of IS Rats

The mRNA expression of *Nf-kb1*, *Nlrp3*, *Caspase1*, and *Il-1β* in brain tissue samples from rats in each group was analyzed using RT-qPCR (Figure 14). PAS840 administration effectively reduced the expression of inflammasome pathway genes (*p* < 0.05). The expression of the neurotrophic factors *Vgf* and *Bdnf*, as well as the vascular endothelial growth factors *Erbb4* and the neuregulin *Nrg1*, were also effectively increased (*p* < 0.01). In conclusion, when IS occurs, the administration of PAS840 can effectively inhibit the inflammasome pathway, reduce the occurrence and development of brain inflammation, activate neurotrophic and vascular endothelial growth factor pathways, protect brain nerve cells, and mitigate injury to the BBB.

#### 3.3.7. Effect of PAS840 on NLRP3 and NF-κB in IS Rats

Immunohistochemical analysis was conducted to determine the expression of NLRP3 and NF-κB in the infarct area of the cortex in the Sham, Model, and high-dose PAS840 groups (*p* < 0.01) (Figure 14). It was found that the administration of PAS840 could effectively reduce the positive expression levels of NLRP3 and NF-κB in the cortical tissues. This was highly consistent with the RT-qPCR results. These results indicated that the administration of PAS840 effectively reduced brain inflammation after the occurrence of IS and slowed the progression of IS.

## 4. Discussion

IS represents a human condition with limited treatment options, an extremely narrow therapeutic window, and extremely high morbidity and mortality [35]. The pathogenesis of IS is very complex, and currently, it includes various theories such as brain nerve inflammation, calcium overload, and ferroptosis; however, they all point to the death of neuronal cells caused by intense stimulation injury, which leads to local inflammation in the injured brain region [36]. This localized brain inflammation exacerbates secondary brain injury by aggravating BBB injury, oxidative stress, and brain edema and directly induces neuronal cell death [37]. In addition to local inflammation in the brain regions, increasing evidence shows that the inflammatory response following stroke extends throughout the brain, leading to complex secondary conditions [38]. Thus, for high-risk populations of the disease, reducing inflammation and oxidative stress damage to thereby mitigate sudden disease-related injuries, and minimizing disease-associated complications and sequelae, represent key focuses for drug development targeting this condition.

Microglia are intrinsic immune effector cells of the central nervous system (CNS) that play a crucial role in neuroinflammation. In acute neurodegenerative diseases (stroke, cerebral hypoxia, and brain trauma), the phenotype of microglia changes, inflammatory mediators are released, and a transformation from the early anti-inflammatory phenotype (M2) to the pro-inflammatory phenotype (M1) occurs, which causes neuroinflammation and ultimately leads to varying degrees of inflammatory responses [39]. M1 phenotype cells secrete various pro-inflammatory factors, such as nitric oxide (NO), reactive oxygen species (ROS), tumor necrosis factor-α (TNF-α), IL-1β, IL-6, and chemokines, etc. [40]. Intense oxidative stress injury mediates the activation of NF-κB and NLRP3 inflammasomes and stimulates the activation of caspase-1 and IL-1β. Therefore, controlling the oxidative stress response and the cascade reaction of NLRP3 inflammasomes has become key to the treatment of IS.

Related studies have shown that PAE exhibits favorable pharmacological effects, such as anti-inflammation, anti-oxidation, the promotion of tissue repair, the reduction of edema, and the promotion of angiogenesis in various injury-related diseases [41], which are highly consistent with IS treatment goals. This study predicted the effects of IS using network pharmacology and combined in vitro experiments to construct a BV-2 cell OGD/R injury model and a rat brain tMCAO injury model to detect the activity of oxidative stress injury and inflammasome pathways induced by IS. We investigated the effects of PAS840 on IS-mediated cerebral cell inflammatory responses and pyroptosis to elucidate the possible mechanisms by which PAS840 treats IS.

LC-MS/MS and peptidomics analyses revealed that PAS840 contained abundant material components. Through network pharmacology prediction analysis and molecular docking, we screened out 93 key therapeutic targets, among which classical inflammatory signal targets such as NOS, MAPK, Caspase, IL-1β, and IL-6 had relatively high correlation scores. These target proteins suggest that inhibiting the body’s oxidative stress response and inflammation and enhancing the body’s resistance to stimulation-induced apoptosis are the keys to the treatment of IS. In subsequent in vitro cell experiments, PAS840 effectively inhibited the levels of ROS, MDA, and NOS and increased SOD activity, thereby alleviating the oxidative stress caused by OGD/R. It could effectively inhibit the activation of the NF-κB/NLRP3/Caspase-1/IL-1β inflammasome pathway, and reduce the inflammatory response and pyroptosis of BV-2 cells caused by OGD/R injury.

By comparing the core therapeutic targets predicted by network pharmacology with the differential genes between the PAS840 and OGD/R groups in transcriptomics, a high degree of overlap was observed, indicating that the predictions based on network pharmacology in this study possessed a high degree of credibility. A further analysis of the core therapeutic targets and differentially expressed genes revealed that both network pharmacology and transcriptome enrichment analysis results demonstrated PAS840’s positive regulatory role in enhancing cellular resistance to stimulation-induced injury and inflammation, regulating neural signal conduction, and promoting anti-apoptotic processes. It also participates in the treatment of various diseases, including immune dysregulation and toxoplasmosis. Notably, signaling pathways, such as neural receptor–ligand engagement, calcium channels, and neural signal propagation, which are intimately associated with a myriad of neurological pathologies, demonstrate pronounced involvement in these conditions and show therapeutic efficacy.

Microvascular endothelial cells form the structural basis of the BBB. The BBB strictly controls the exchange of nutrients and metabolites between the blood and brain tissue and plays a vital role in maintaining the homeostasis of the cerebral microenvironment and normal neural functions [42]. BBB disruption is a major pathological manifestation of IS and continuously aggravates the disease course at different stages of cerebral ischemia. These processes include matrix metalloproteinase (MMP) activation, inflammatory expression, vesicular transport, cellular oxidative stress, and cytoskeleton disruption, which further undermine the maintenance of cell stability and nutrient and metabolic transport in the CNS, thereby exacerbating ischemic brain injury and tissue-type plasminogen activator injury [42]. Therefore, BBB injury and leakage caused by IS and the resulting series of complications are important causes of aggravated IS injury and impeded recovery from the disease course [43]. Thus, in BBB repair, inhibiting oxidative stress and brain inflammation, reconstructing the microvascular system, and activating neurotrophic factors are important strategies for the modern medical treatment of IS and for reducing the occurrence of complications [44].

In in vivo animal experiments, the results of HE staining showed that the administration of PAS840 could significantly reduce brain tissue damage. We further explored the in vivo expression of the EB dye solution and its impact on IS through EB staining. We found that after the administration of PAS840, the physiological state of the rats in the treatment group was significantly better than that in the Model group, and the EB staining area and EB content in the brain tissue decreased significantly. Through qPCR experiments and immunohistochemistry experiments, it was found that PAS840 could effectively reduce the expression of the NF-κB/NLRP3/Caspase-1/IL-1β inflammasome pathway and simultaneously activate the BDNF/VGF/NGR1/Erbb4 neurotrophic factors as well as vascular endothelial growth factor. The results of both in vivo and in vitro experiments indicated that PAS840 could effectively inhibit the expression of the inflammasome pathway. This is highly consistent with the results of in vitro cell experiments, and the inhibition of NLRP3 expression, alleviation of oxidative stress, and protection of neuronal cells and the BBB are key factors in modern IS treatment.

Research findings demonstrate that PAS840 mitigates the inflammatory response and oxidative stress levels in BV-2 cells after OGD/R injury, enhances cell viability and migratory capacity, and decreases inflammation and oxidative stress levels in rat brain tissue after tMCAO. PAS840 also alleviates BBB injury and reduces neuronal cell pyroptosis by suppressing the NF-κB/NLRP3/Caspase-1/IL-1β inflammasome pathway and activating the expression of BDNF/VGF/NGR1/Erbb4 neurotrophic factors and vascular endothelial growth factor. This indicates that the pharmacological effects in question likely represent the pivotal mechanism by which PAS840 exerts its therapeutic benefits in the treatment of IS.

Although this study indicated that PAS840 exerts neuroprotective and therapeutic effects against IS-induced brain injury in cellular and animal models, and also indicated its protective role against IS occurrence, it is limited by its reliance on BV-2 microglial cell analysis and exclusive focus on neuroinflammation, thus lacking an exploration of other pathological sites of IS. While immortalized cells enhance experimental reproducibility, model expansion is required, including neuronal cells, human cerebral vascular endothelial cells, and organoid co-culture systems, to validate therapeutic feasibility.

In future work, we will investigate and evaluate the effects of preoperative and postoperative administration regimens of this drug on IS, conduct in vivo experiments in female rats to explore its efficacy, and complete acute toxicity tests and long-term safety assessments. Finally, the integration of multi-omics approaches will be used to decipher mechanisms and optimize synergies with existing therapies, thereby advancing the drug’s development as a preventive and surgical adjuvant for further research and clinical translation.

## 5. Conclusions

This study predicts the therapeutic effects of PAS840, an extract from *Periplaneta americana* (L.), through network pharmacology combined with in vitro and in vivo studies. The results indicate that PAS840 can inhibit the expression of the NLRP3/Caspase-1/IL-1β inflammasome pathway, and activate the BDNF/VGF/NGR1/Erbb4 neurotrophic factor and vascular endothelial growth factor, thereby alleviating the inflammatory response in the brain following IS, reducing oxidative stress levels, enhancing neuronal cell viability, and effectively protecting and repairing the blood–brain barrier. Thus, PAS840 is a potential therapeutic agent for IS and warrants further research and development as a preventive drug for high-risk populations and as a surgical adjuvant.

## Figures and Tables

**Figure 1 biology-14-00589-f001:**
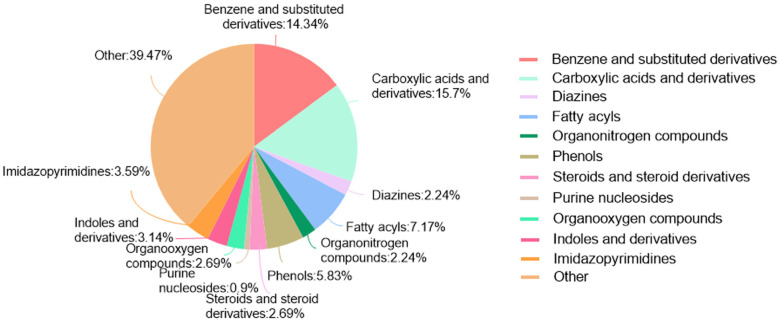
Pie chart of the percentage of small-molecule compounds in PAS840.

**Figure 2 biology-14-00589-f002:**
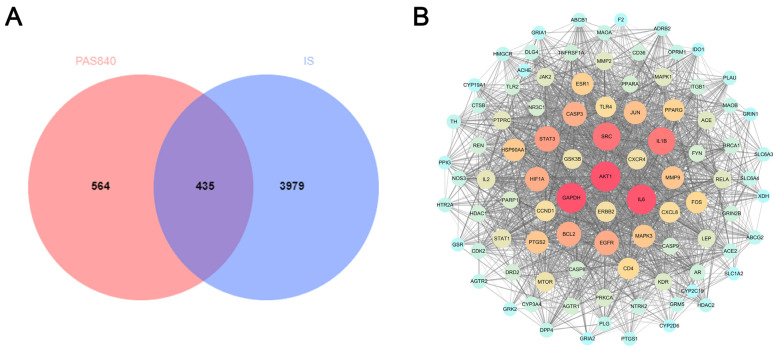
Pharmacological target analysis of PAS840 component networks. (**A**) Intersecting protein Venn diagram. (**B**) Intersecting protein PPI network diagram. The darker the node, the greater the degree value, and the darker and denser the line, the greater the protein correlation.

**Figure 3 biology-14-00589-f003:**
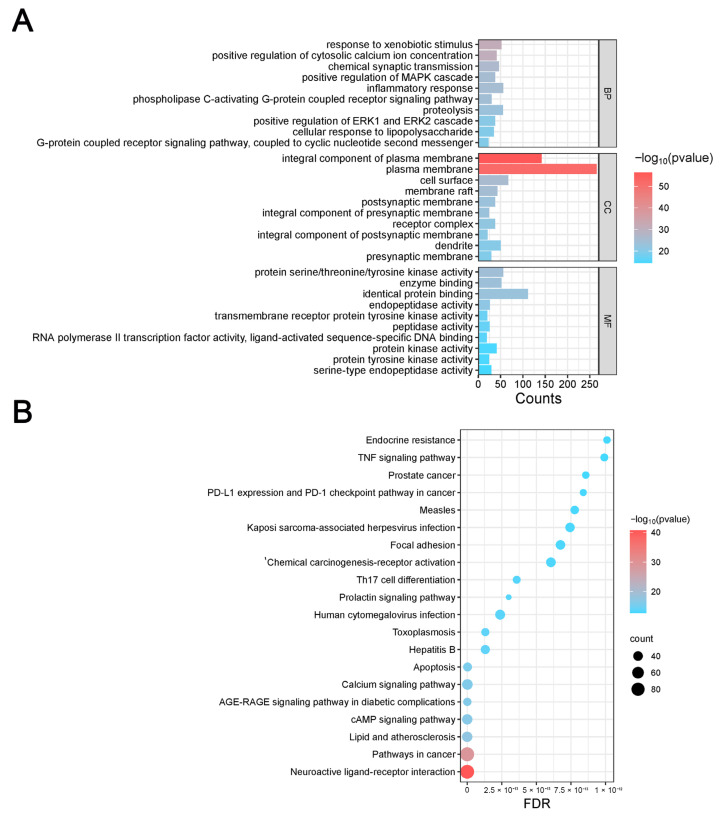
Enrichment analysis of PAS840 in IS. (**A**) Bar graph of GO enrichment analysis; (**B**) bubble network diagram of KEGG enrichment analysis.

**Figure 4 biology-14-00589-f004:**
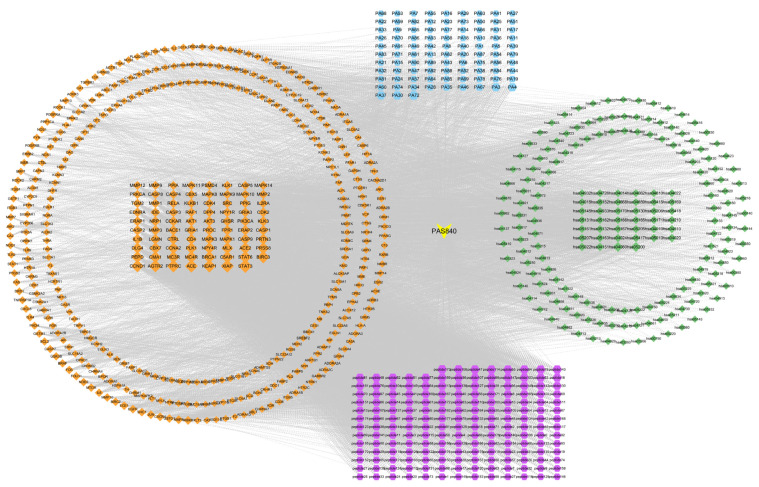
PAS840–disease–pathway interaction network map. PAS840 samples are shown in yellow, disease targets in orange, small-molecule compounds in blue, peptides in purple, and KEGG pathways in green; the larger the degree value of the correlation of each color block, the larger the size of the color block.

**Figure 5 biology-14-00589-f005:**
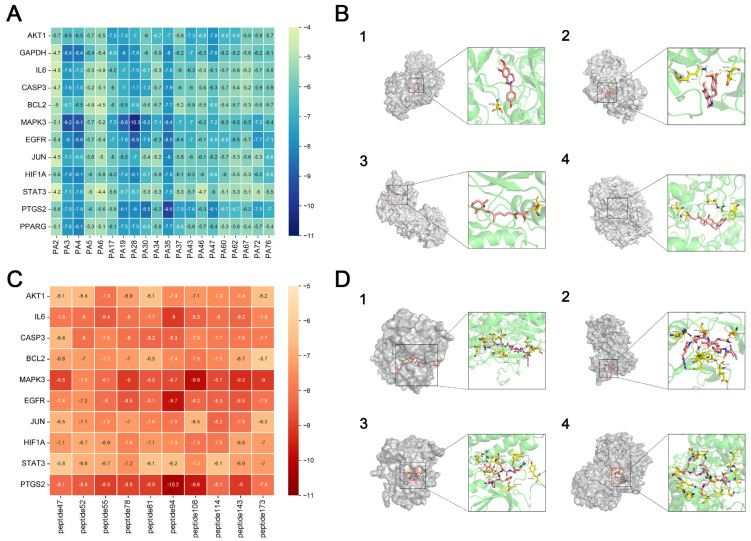
Visual display of molecular docking results. (**A**) Heatmap of molecular docking scores of small-molecule compounds; (**B1**) MAKP3 docked with (S)-N-methylcoclaurine; (**B2**) MAPK3 docked with (S)-reticuline; (**B3**) MAPK3 docked with all-trans-retinoic; (**B4**) PTGS2 docked with deoxycholic; (**C**) heatmap of molecular docking scores of peptides; (**D1**) MAKP3 docked with FDQLGR; (**D2**) MAKP3 docked with ATFDQLGR; (**D3**) EGFR docked with TPFYLR; and (**D4**) PTGS2 docked with FDQLGR.

**Figure 6 biology-14-00589-f006:**
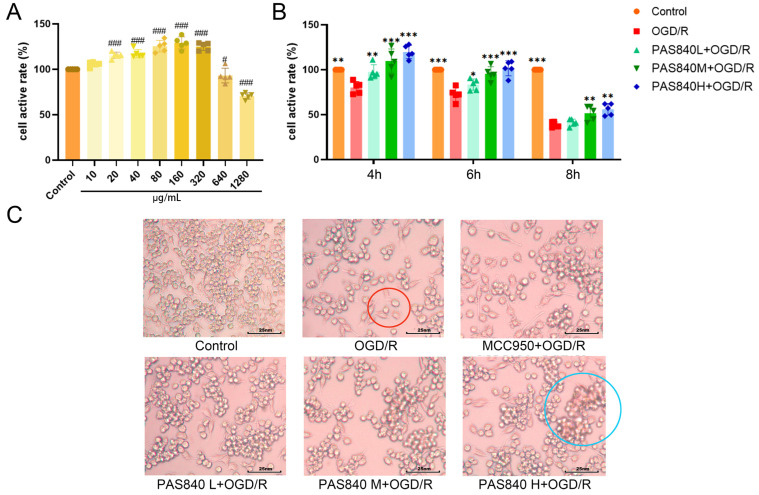
Cell viability and state. (**A**) Effect of PAS840 on cell viability (*n* = 5). (**B**) Cell viability at three different OGD/R injury times (*n* = 5). (**C**) Cell state after OGD/R injury (*n* = 3). Red circles indicate differentiated cells; blue circles indicate high-activity cell clusters. # *p* < 0.05, and ### *p* < 0.001 vs. Control; * *p* < 0.05, ** *p* < 0.01, and *** *p* < 0.001 vs. OGD/R group.

**Figure 7 biology-14-00589-f007:**
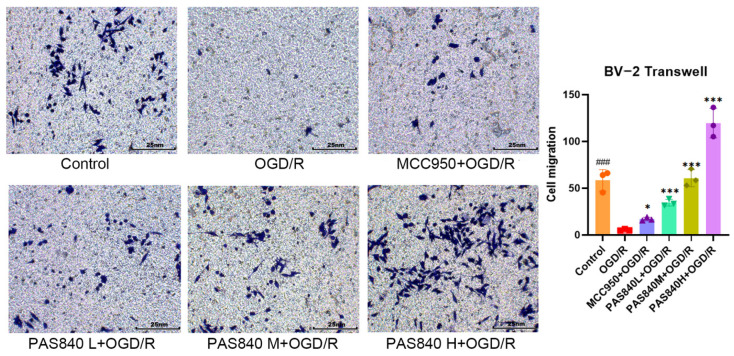
Number of cell migrations (*n* = 3). ### *p* < 0.001 vs. Control; * *p* < 0.05, and *** *p* < 0.001 vs. OGD/R group.

**Figure 8 biology-14-00589-f008:**
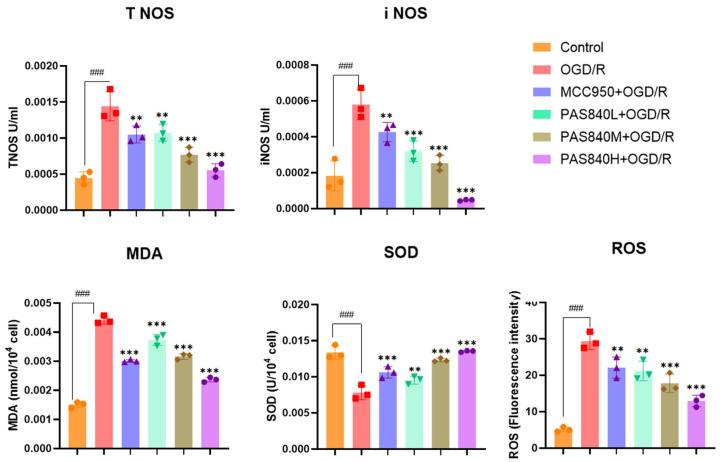
The effect of PAS840 on the expression of MDA, SOD, T NOS, i NOS, and ROS in injured cells (*n* = 3). ### *p* < 0.001 vs. Control; ** *p* < 0.01, and *** *p* < 0.001 vs. OGD/R group.

**Figure 9 biology-14-00589-f009:**
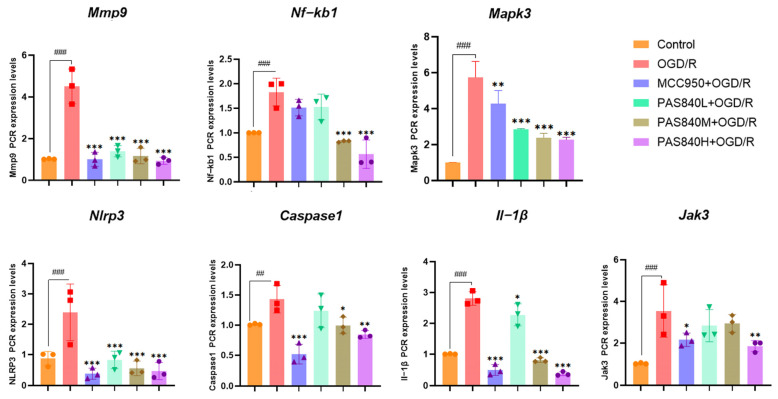
Effect of PAS840 on the mRNA expression of *Mmp9*, *Nf-kb1*, *Nlrp3*, *Caspase1*, *Il-1β*, *Jak3*, and *Mapk3* in injured cells (*n* = 3). ## *p* < 0.01, and ### *p* < 0.001 vs. Control; * *p* < 0.05, ** *p* < 0.01, and *** *p* < 0.001 vs. OGD/R group.

**Figure 10 biology-14-00589-f010:**
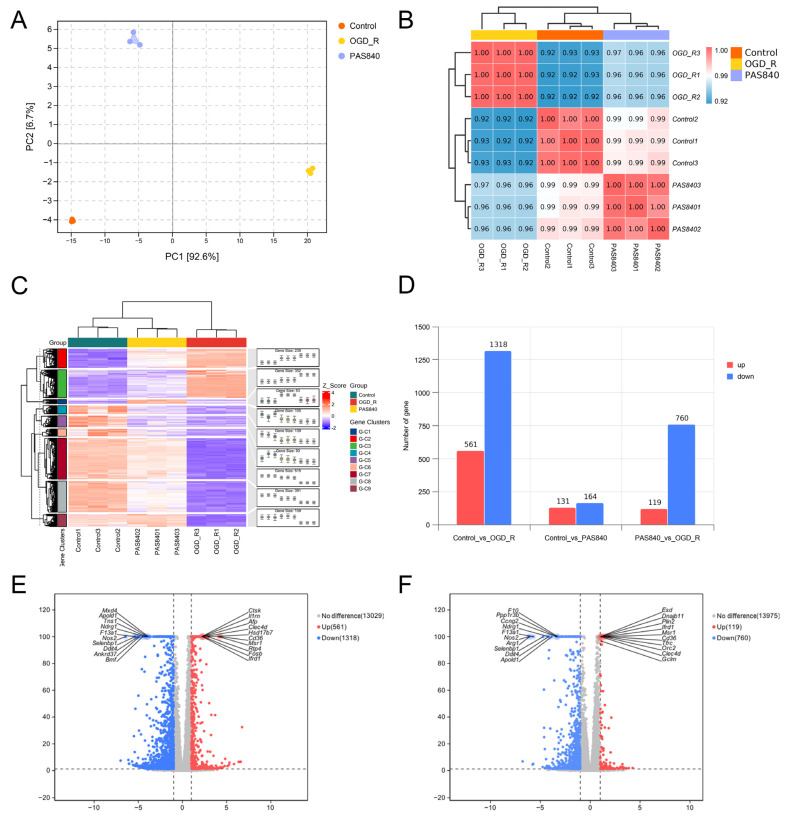
Effect of PAS840 on RNA expression after administration. (**A**) PCA analysis diagram. (**B**) Heat maps for correlation analysis. (**C**) Heat maps for cluster analysis. (**D**) Differential gene expression histogram. (**E**) Control vs. OGD/R differential gene expression volcano plot. (**F**) PAS840H vs. OGD/R differential gene expression volcano plot.

**Figure 11 biology-14-00589-f011:**
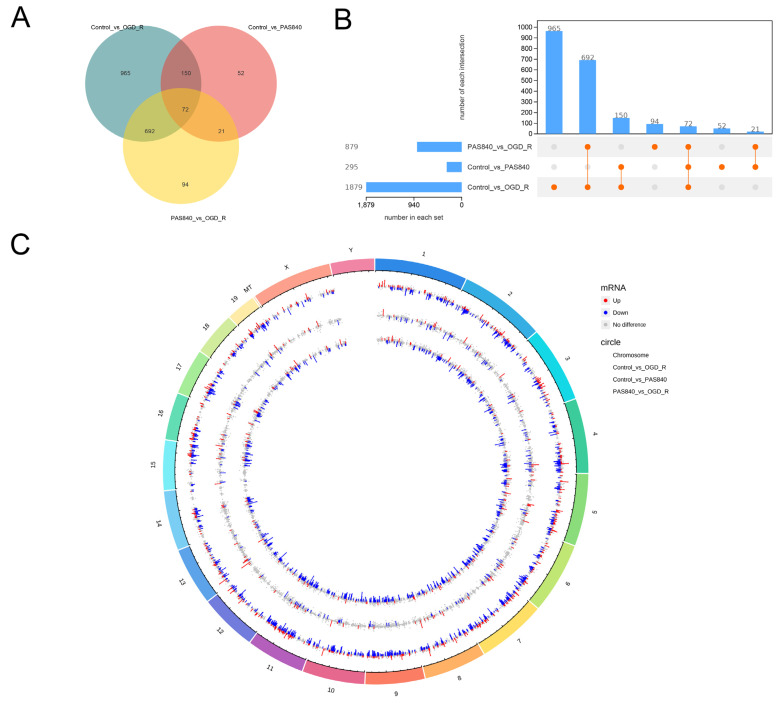
Analysis of RNA differences among groups. (**A**) A Venn diagram illustrating differential gene expression. (**B**) An UpSet plot depicting differential gene expression. (**C**) A genomic map displaying differential gene expression. The outermost circle of the genomic map represents chromosome bands, with red and blue bars indicating the log2FoldChange values for upregulated and downregulated genes, respectively.

**Figure 12 biology-14-00589-f012:**
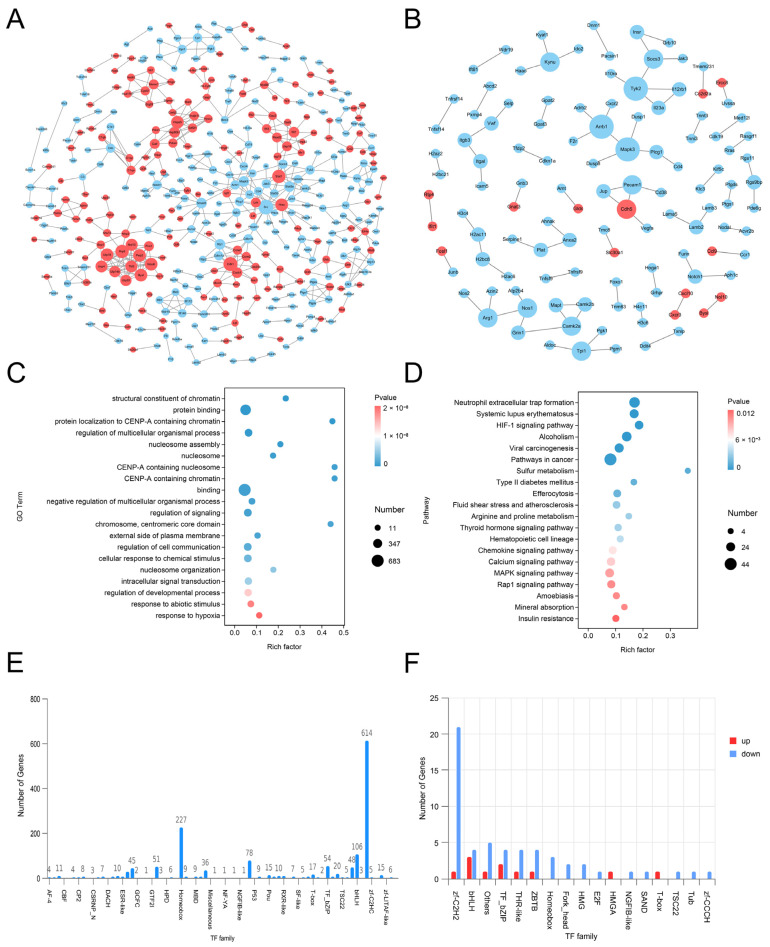
Differential gene analysis. (**A**) The network diagram of differential gene protein interaction between the Control group vs. OGD/R group. (**B**) The network diagram of differential gene protein interaction between the PAS840 treatment group vs. OGD/R group. (**C**) GO bubble plot. (**D**) KEGG bubble plot. (**E**) Histogram of transcription factor family distribution. (**F**) Histogram of differential expression of transcription factors between PAS840 treatment group and OGD/R group. In the network diagram, red represents upregulated genes, blue represents downregulated genes, and the larger the circle, the higher the score of the gene.

**Figure 13 biology-14-00589-f013:**
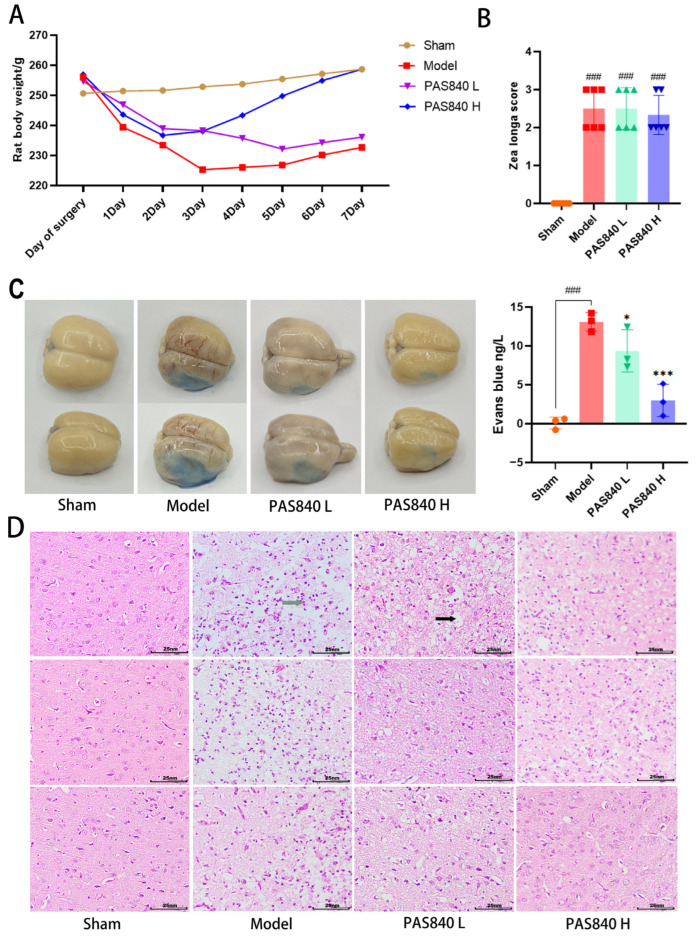
Effects of PAS840 on physiological indexes of IS rats. (**A**) Dynamic monitoring of body weight change (*n* = 6). (**B**) Zea Longa Neurological Function Score (*n* = 6). (**C**) Evans blue staining (*n* = 3). (**D**) HE staining of brain tissue sections (*n =* 3). ### *p* < 0.001 vs. Sham; * *p* < 0.05, and *** *p* < 0.001 vs. Model group. 
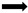
 indicates cellular vacuolation and 
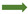
 indicates inflammatory cell invasion.

**Figure 14 biology-14-00589-f014:**
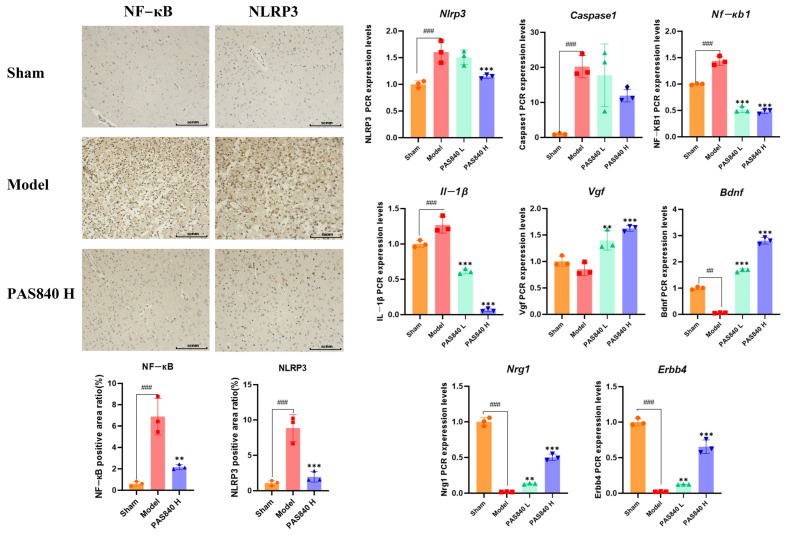
RT-qPCR and immunohistochemical analysis of PAS840 on brain tissue of IS rats (*n* = 3), ## *p* < 0.01, and ### *p* < 0.001 vs. Sham group; * *p* < 0.05, ** *p* < 0.01, and *** *p* < 0.001 vs. Model group.

**Table 1 biology-14-00589-t001:** PAS840 small-molecule gradient elution program.

Time		Velocity of Flow
POS	0–1 min	8% B2
	1–8 min	8–98% B2
	8–10 min	98% B2
	10–10.1 min	98–8%B2
	10.1–12 min	8% B2
NEG	0–1 min	8% B3
	1–8 min	8–98% B3
	8–10 min	98% B3
	10–10.1 min	98–8% B3
	10.1–12 min	8% B3

**Table 2 biology-14-00589-t002:** PAS840 peptide chromatographic gradient.

Time	Proportion of Mobile Phase B (%)
0–7 min	8% B
7–55 min	12% B
55–65 min	30% B
65–66 min	40% B
66–80 min	95% B
80 min	95% B

**Table 3 biology-14-00589-t003:** Primer sequence (mouse).

Name	Primer Direction	Primer Sequence (5′-3′)
*Nf-kb1*	Forward	CTGAGAAGGAAACTGAAGGTGAAGGG
	Reverse	GAGGTGGATGATGGCTAAGTGTAAGAC
*Caspase1*	Forward	GCCGTGGAGAGAAACAAGGAGTG
	Reverse	TCCAAGTCACAAGACCAGGCATATTC
*Il-1* *β*	Forward	TTCAGGCAGGCAGTATCACTCATTG
	Reverse	TGTCGTTGCTTGGTTCTCCTTGTAC
*Nlrp3*	Forward	TCCGAAAGAAACTGCTGCCCAAG
	Reverse	GTCAGAGAAGAGATGCTCCTCAATGC
*Mmp9*	Forward	CGCCACCACAGCCAACTATGAC
	Reverse	GATACTGGATGCCGTCTATGTCGTC
*Jak3*	Forward	ATAGGCATCCGAGACATCCTCAGAG
	Reverse	AAGGTCGCAGGTGGTGTTGATAAG
*Mapk3*	Forward	CAGAGCCTGCGGTTGGTGATG
	Reverse	GCGGGAGAAGATGTTGTCAGATAGG
*β-actin*	Forward	TGCTGTCCCTGTATGCCTCTGG
	Reverse	ACCGCTCGTTGCCAATAGTGATG

**Table 4 biology-14-00589-t004:** Primer sequence (rat).

Name	Primer Direction	Primer Sequence (5′-3′)
*Nf-kb1*	Forward	GCGGTTACGGGAGATGTGAAGATG
	Reverse	AGTGCTGCCTTGCTGTTCTTGAG
*Caspase1*	Forward	GCCGTGGAGAGAAACAAGGAGTG
	Reverse	GGTCACCCTTTCAGTGGTTGGC
*Il-1* *β*	Forward	TCTGTGACTCGTGGGATGATGAC
	Reverse	TTTGTCGTTGCTTGTCTCTCCTTG
*Nlrp3*	Forward	CTGCGGACTGACCCATCAATGC
	Reverse	ACCAATGCGAGATCCTGACAACAC
*Vgf*	Forward	GGATGAGGAGGCGGCAGAGG
	Reverse	CAGGTGGAGTTTGGTGGACAGTTC
*Bdnf*	Forward	GTGATCGAAGAGCTGCTGGATGAG
	Reverse	GCTGTGACCCACTCGCTAATACTG
*Nrg1*	Forward	GAGAACAGCAGGCACAGCAGTC
	Reverse	GGCGTCACAAGCAGCAGAGG
*Erbb4*	Forward	CGAGCAGAACTGGATGAAGAAGGC
	Reverse	AAGGCGTTGGTGAAGGTGTTGAG
*β-actin*	Forward	GCTGTGCTATGTTGCCCTAGACTTC
	Reverse	GGAACCGCTCATTGCCGATAGTG

**Table 5 biology-14-00589-t005:** Top 20 small-molecule compounds with highest relevance.

Number	Name	ID	Molecular Formula	Structures
1	(S)-2-Propylpiperidine	PA2	C8H17N	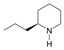
2	(S)-N-Methylcoclaurine	PA3	C18H21NO3	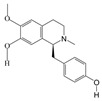
3	(S)-Reticuline	PA4	C19H23NO4	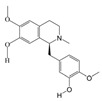
4	1,2,3-Trihydroxybenzene	PA5	C6H3(OH)3C6H6O3	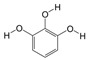
5	1,3-Benzenediol	PA6	C6H4(OH)2C6H6O2	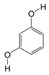
6	3-Methyl-L-tyrosine	PA17	C10H13NO3	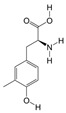
7	4-(3-Methylbut-2-enyl)-L-tryptophan	PA19	C16H20N2O2	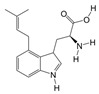
8	all-trans-Retinoic acid	PA28	C20H28O2	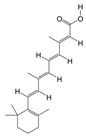
9	Capsaicin	PA30	C18H27NO3	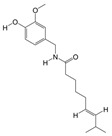
10	D-Phenylalanine	PA34	C9H11NO2	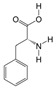
11	Deoxycholic acid	PA35	C24H40O4	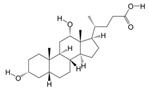
12	Dibutyl phthalate	PA37	C16H22O4	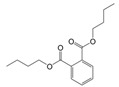
13	Formylanthranilic acid	PA43	C8H7NO3	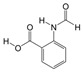
14	Hordenine	PA46	C10H15NO	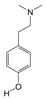
15	Hydralazine	PA47	C8H8N4	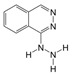
16	L-Phenylalanine	PA60	C9H11NO2	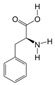
17	L-Tyrosine	PA62	C9H11NO3	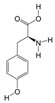
18	N-Acetylleucine	PA67	C8H15NO3	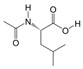
19	PGA1	PA72	C20H32O4	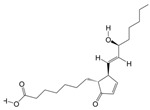
20	Phenylpyruvic acid	PA76	C9H8O3	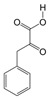

**Table 6 biology-14-00589-t006:** Peptides used for molecular docking.

Number	Peptide	ID	Charge
1	KDEGLNGFYK	peptide47	0
2	LVEFPK	peptide52	0
3	VGLEQYVPPK	peptide55	0
4	FDLVSR	peptide78	0
5	AGNALVEFPK	peptide81	0
6	TPFYLR	peptide94	1
7	FDQLGR	peptide108	0
8	TFYNELR	peptide114	0
9	ATFDQLGR	peptide143	0
10	LFLDKLR	peptide173	1

## Data Availability

All of the data are uploaded into the Mendeley Data database (https://data.mendeley.com/datasets/gj8h4yssdd/1, accessed on 26 November 2024); the data set name is the same as the article title, DOI: 10.17632/gj8h4yssdd.1.

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
