# Peer review of "Periplaneta americana (L.) Extract PAS840 Promotes Ischemic Stroke Recovery by Inhibiting Inflammasome Activation"

_biology, 2025, doi:10.3390/biology14060589_

Round 1

Reviewer 1 Report (Previous Reviewer 2)

Comments and Suggestions for Authors

Overall, the authors have made significant improvements to the manuscript, addressing many of the previous concerns. However, there are still areas that require further refinement. Specifically, the Methods section should include a more detailed description of the experimental procedures to enhance reproducibility. This study provides valuable insights and has the potential to contribute significantly to future research in this field. This manuscript can be accepted in present format also.

Author Response

We are deeply grateful for your constructive feedback and honored by your recognition of our work. In response to your suggestions, we have conducted a comprehensive revision of the manuscript, including the following enhancements:

Data Visualization Optimization: Restructured graphical presentations to improve interpretability and align with journal formatting standards.

Methodological Precision: Refined descriptions of critical experimental parameters, including cell passage numbers and rat age (specified in weeks).

Technical Transparency: Expanded reporting of procedural details to ensure full reproducibility.

Your endorsement has been instrumental in elevating the scientific rigor of this study. Should you identify any additional areas for improvement, we earnestly welcome your further guidance to help us perfect this manuscript.

Reviewer 2 Report (New Reviewer)

Comments and Suggestions for Authors

The manuscript presents an interesting study investigating the therapeutic effects of PAS840 in ischemic stroke using both in vitro and in vivo models. While the findings are promising, some queries need to be addressed while revising the manuscript.

  • Was PAS840 diluted in complete culture medium during treatment? If so, serum components might affect its bioavailability or activity. Justify it.
  • Ischemic Stroke shows sex-specific effects. Please justify the use of only male rats and clarify if sex differences were considered.
  • The sample size used for in vivo (n = 6 each group; control=3, treatment=3) is quite limited. Was a statistical power calculation performed to justify the group size?
  • The rationale for administering PAS840 both before and after tMCAO induction is unclear. Justify it.
  • Please justify the choice of 6 hours after the final drug administration for tissue collection. Was this time point based on pharmacodynamic considerations or previous findings?
  • Please specify the vehicle used for PAS840 administration and confirm whether the same vehicle was used in the model and sham groups to ensure a valid comparison.
  • Please clarify how successful tMCAO induction was confirmed (e.g., neurological scoring or blood flow monitoring).
  • Line 370, “Take the remaining rat brain tissue” Please specify which region of the brain tissue was used for qPCR analysis, as gene expression can vary across brain regions after ischemic injury.
  • The manuscript lacks a direct assessment of infarct damage. It is recommended to include infarct analysis, such as TTC or cresyl violet staining, to evaluate the neuroprotective effects of PAS840.
  • Ischemic stroke results in both acute and chronic behavioral deficits. However, no behavioral assessments were performed in this study. It is recommended to include at least acute behavioral evaluations to better assess the functional impact of PAS840 treatment. Additionally, the author could include infarct analysis, such as TTC or cresyl violet staining, to evaluate the neuroprotective effects of PAS840.
  • Please clarify which OGD/R condition (duration of OGD) was used for the experiments shown in Fig. 6c and for subsequent experiments.
  • In Figure 6B, it appears that all groups are compared to the control group. However, the OGD/R group does not show any significance markers (e.g., *), despite an apparent reduction in viability. Additionally, for the 8-hour OGD group, the PAS840+L treatment lacks significance, and the M and H groups show only "**" despite visibly large differences. Please clarify the statistical comparisons, including which groups were compared and whether proper post-hoc tests were applied.
  • In Figures 6–9 and 13-14, please include individual data points (e.g., as scatter or bubble overlays) on each bar graph to better illustrate sample variability and distribution.
  • The study relies heavily on BV-2 microglial cells for in vitro analysis. Since BV-2 cells are an immortalized line and may not fully recapitulate primary microglial responses, could the authors justify their use and address potential limitations in translatability?

Minor

  • In the abstract, Line 31, “Mouse microglial cells (BV-2) cells” remove one cells.
  • Please ensure consistent abbreviation usage throughout the manuscript. Once a term such as "ischemic stroke (IS), oxygen glucose deprivation/reoxygenation (OGD/R) " has been introduced and abbreviated, repeating the full term in subsequent mentions is unnecessary. Using only the abbreviation (e.g., IS, OGD/R) improves clarity and avoids redundancy.
  • Please include the complete medium composition and BV-2 cell passage number, as these details are important for reproducibility.
  • Mention the age of the rats used for the experiments.
  • Line 286, “Real-time quantitative polymerase chain reaction” correct the font size.
  • Line 292, 2-ΔΔCT change to 2-ΔΔCT
  • Line 295, method “2.4.6. Transcriptomic sequencing” is at the wrong position. It should be after Table 3.

Author Response

We deeply appreciate your evaluation of our manuscript, which serves as a significant motivation for our research progress. Through careful review of your insightful comments, we have gained a thorough understanding of the deficiencies in our manuscript. In response, we have carefully revised the content, with particular emphasis on supplementing the preliminary theoretical foundation and experimental basis of this study. Below are our detailed responses to your feedback for your review.

Q1. Was PAS840 diluted in complete culture medium during treatment? If so, serum components might affect its bioavailability or activity. Justify it.

Response: Thank you for raising this question, which encourages us to refine our experimental approach. In modern cell experiments, many compounds require PBS or DMSO for dissolution due to solubility limitations. However, PAS840 in this study is fully soluble in complete medium, eliminating the influence of PBS or DMSO. During oxygen-glucose deprivation (OGD), we used serum- and glucose-free medium to mimic ischemic conditions. Post-OGD, serum and glucose were reintroduced to simulate nutrient restoration in the brain. We believe that the concurrent presence of serum and PAS840 better reflects the physiological environment for drug efficacy during recovery.

Q2. Ischemic Stroke shows sex-specific effects. Please justify the use of only male rats and clarify if sex differences were considered.

Response: Thank you for your questions and suggestions. In our preliminary research on ischemic stroke (IS), we found that the incidence rate of this disease is significantly higher in males than in females. Moreover, male mice or rats are predominantly used in mainstream animal experiments for IS studies [1]. Relevant literature has demonstrated that estrogen (e.g., estradiol) plays a remarkably significant role in IS treatment [2-4]. Furthermore, our previous studies revealed that female rats exhibit much faster neurofunctional recovery post-IS compared to males. Typically, 24 hours after model establishment, 50% of the female rats in the model group scored 1 point on the Zea Longa scale. Therefore, we believe that the investigation of PAS840's effects on IS in female rats warrants a separate research project. We are currently planning to initiate this study in the third quarter of this year.

Q3: The sample size used for in vivo (n = 6 each group; control=3, treatment=3) is quite limited. Was a statistical power calculation performed to justify the group size?

Response: In our experiments, to ensure sample uniformity and enhance credibility, all animal models were established solely by the first author of the manuscript within a single day. Due to the relatively high technical difficulty of the modeling procedure, each rat surgery required approximately 20–25 minutes, which significantly limited our biological sample size. Our experimental design originally included 9 rats per group, and we completed modeling for 36 rats on the same day. However, due to unsuccessful TTC staining, 3 samples from each group were excluded, resulting in only 6 samples per group being available for final analysis. To address this, we have supplemented the unsuccessful TTC staining results in Q9 of this response letter. Additionally, to comply with the animal welfare requirements of our institution’s Animal Welfare Committee, we selected the statistically significant minimum sample size for this study. We sincerely hope for your understanding regarding the limitations in our experimental design and technical challenges.

Q4. The rationale for administering PAS840 both before and after tMCAO induction is unclear. Justify it.

Response: Thank you for raising this concern. The rationale for administering PAS840 prior to tMCAO was to investigate whether it could mitigate IS-induced injury during the onset of IS. While this approach may lack rigorous scientific justification, it did demonstrate a degree of protective effect against IS-related damage. We sincerely appreciate your critical feedback, which has highlighted opportunities to improve the rigor and completeness of our experimental design.

Q5. Please justify the choice of 6 hours after the final drug administration for tissue collection. Was this time point based on pharmacodynamic considerations or previous findings?

Response: Thank you for your question. The decision to collect brain tissues 6 hours after the final administration was based on our preliminary assessment of peak drug concentration in rat cerebrospinal fluid. This time frame ensures that the drug distribution and metabolism in brain tissues reach a relative peak, aligning with the kinetic profile of these biological processes. While the 6-hour interval serves as a relatively optimal observation window for drug effects in this study, whether longer administration-to-sampling intervals could better elucidate adaptive regulatory mechanisms will be a critical focus of subsequent research.

Q6. Please specify the vehicle used for PAS840 administration and confirm whether the same vehicle was used in the model and sham groups to ensure a valid comparison.

Response: In our in vivo experiments, all drugs were dissolved in saline and administered via oral gavage. Both the sham group and model group also received equivalent volumes of saline. We acknowledge that this description lacked sufficient detail in our original manuscript. We have now revised and supplemented the methodology section (Lines 338-340) to clarify these procedures. We sincerely appreciate your diligence in identifying this oversight, which has significantly improved the rigor of our manuscript.

Q7. Please clarify how successful tMCAO induction was confirmed (e.g., neurological scoring or blood flow monitoring).

Response: Thank you for your suggestions. During the experiments, we performed Zea Longa scoring and cerebral blood flow (CBF) monitoring in all rat groups. However, due to the thick skulls of the rats, our laser speckle contrast imaging (LSCI) system failed to penetrate the skull, resulting in unsuccessful CBF measurements. Regarding the Zea Longa scores, in our previously published article, reviewers advised that neurological scoring should only serve to confirm model success and cannot directly demonstrate pharmacological efficacy, recommending its removal. Consequently, we omitted the Zea Longa scores in this manuscript. In response to your feedback, we have now supplemented the Zea Longa scoring data in Sections 2.5.4 and 3.3.2 of the revised manuscript to validate the successful establishment of the tMCAO model.

Q8. Line 370, “Take the remaining rat brain tissue” Please specify which region of the brain tissue was used for qPCR analysis, as gene expression can vary across brain regions after ischemic injury.

Response: We sincerely appreciate your inquiry. The brain tissues analyzed in our experiments were specifically obtained from the cortical region of the infarct area in the right cerebral hemisphere (the hemisphere subjected to the surgical procedure). We have provided additional clarification regarding this methodological detail in Line 381 of the revised manuscript's methodology section. Thank you for highlighting this point, as it has strengthened the precision of our documentation.

Q9. The manuscript lacks a direct assessment of infarct damage. It is recommended to include infarct analysis, such as TTC or cresyl violet staining, to evaluate the neuroprotective effects of PAS840.

Response: Thank you for raising this critical question. Our experimental design did incorporate TTC staining for lesion assessment. However, technical challenges during the staining process caused partial tissue section damage and indistinct staining boundaries, which precluded precise quantification of the infarct area. Despite this limitation, the images visually demonstrate that PAS840 significantly reduced the infarct size. To ensure transparency, we have included the original TTC staining images as supplementary material for reader access and evaluation. Additionally, the complete TTC images are provided in the PDF response file accompanying this revision. We sincerely apologize for the shortcomings arising from these technical difficulties and deeply appreciate your understanding and constructive feedback.

  •  

Q10. Ischemic stroke results in both acute and chronic behavioral deficits. However, no behavioral assessments were performed in this study. It is recommended to include at least acute behavioral evaluations to better assess the functional impact of PAS840 treatment. Additionally, the author could include infarct analysis, such as TTC or cresyl violet staining, to evaluate the neuroprotective effects of PAS840.

Response: Thank you for your valuable suggestion. We have incorporated the Zea Longa scoring data into the revised manuscript (Sections 2.5.4 and 3.3.2) and included the TTC staining results as supplementary material.

Q11: Please clarify which OGD/R condition (duration of OGD) was used for the experiments shown in Fig. 6c and for subsequent experiments.

Response: Thank you for your critical suggestion. We acknowledge this as a significant oversight in our manuscript preparation and have supplemented the relevant content in Lines 512-514 of the revised manuscript. In this study, we selected a 7-hour OGD/R (oxygen-glucose deprivation/reperfusion) injury duration for subsequent experiments. This decision was based on our rationale that in cellular injury models, a baseline survival rate of 50–60% represents an optimal concentration for modeling agents. Under these conditions, if the tested compound can restore cellular survival to higher levels, it more robustly demonstrates the compound’s therapeutic efficacy.

Q12: In Figure 6B, it appears that all groups are compared to the control group. However, the OGD/R group does not show any significance markers (e.g., *), despite an apparent reduction in viability. Additionally, for the 8-hour OGD group, the PAS840+L treatment lacks significance, and the M and H groups show only "**" despite visibly large differences. Please clarify the statistical comparisons, including which groups were compared and whether proper post-hoc tests were applied.

Response: Thank you for your insightful question. We acknowledge this as another critical oversight in our manuscript preparation. While we conducted statistical comparisons between the control group and the OGD/R group, we regrettably failed to annotate these comparisons in the original figures. We have now comprehensively revised Figure 6B: # denotes comparisons with the control group, while * indicates comparisons with the OGD/R-treated group.

Regarding the 8-hour OGD/R group, the low-dose PAS840 (L) group indeed showed no significant therapeutic effect compared to the OGD/R group. We attribute this primarily to two factors: The 8-hour OGD/R duration induced excessive cellular damage, the PAS840 concentration used had not yet reached its effective threshold under such severe injury conditions. These observations informed our subsequent selection of a 7-hour OGD/R duration for mechanistic studies, and supplementary modifications were made in lines 513-515. Importantly, the dose-dependent increase in cell viability with escalating PAS840 concentrations aligns with the fundamental pharmacological principle of concentration-effect correlation. Therefore, we maintain that these statistical outcomes retain scientific validity for demonstrating therapeutic efficacy.

Q13. In Figures 6–9 and 13-14, please include individual data points (e.g., as scatter or bubble overlays) on each bar graph to better illustrate sample variability and distribution.

Response: Thank you for your valuable suggestion. We have revised the visual presentation of the bar charts in Figures 6–9 and 13–14 to enhance clarity and methodological transparency.

Q14. The study relies heavily on BV-2 microglial cells for in vitro analysis. Since BV-2 cells are an immortalized line and may not fully recapitulate primary microglial responses, could the authors justify their use and address potential limitations in translatability?

Response: We sincerely appreciate the reviewer’s insightful comments. We fully acknowledge the physiological relevance of primary microglia and have carefully considered the potential limitations of the BV-2 cell line in our study design. Our selection of the BV-2 cell line was primarily based on the following rationale:

Scientific Validation: As an internationally recognized microglial model, BV-2 cells exhibit high consistency with primary cells in core phenotypes such as neuroinflammatory responses and phagocytic function [5-7].

Experimental Reproducibility: The homogeneity and stability of BV-2 cells minimize donor variability inherent to primary cultures, ensuring robust mechanistic exploration in vitro.

That said, we fully agree with the reviewer that the immortalized nature of BV-2 cells may introduce discrepancies in metabolic activity, polarization plasticity, and other functional dimensions compared to primary microglia. To address this, our future work will employ advanced models, including organoid co-culture systems and multicellular interaction platforms, to validate the physiological relevance of our findings within brain-mimetic microenvironments.

We are deeply grateful for your constructive critique, which has strengthened both the innovation and methodological transparency of this study. We hope these clarifications adequately address your concerns.

Minor

Q1. In the abstract, line 31: Remove one redundant "cell" from "BV-2 cells (microglial cells)."

Response: It has been modified as required.

Q2. Ensure consistent abbreviation usage throughout the manuscript. Terms like "ischemic stroke (IS)" and "oxygen-glucose deprivation/reoxygenation (OGD/R)" should retain only their abbreviations (IS, OGD/R) after initial definition to enhance clarity and reduce redundancy.

Response: It has been modified as required.

Q3. Include full medium composition and passage numbers for BV-2 cells to ensure reproducibility.

Response: Added in Lines 239–240 of the revised manuscript.

Q4. Specify the age of rats used in experiments.

Response: Updated in Line 329.

Q5. Adjust font size for "quantitative real-time polymerase chain reaction (qPCR)" in Line 286.

Response: It has been modified as required.

Q6. Revise "2-ΔΔCT" to "2−ΔΔCT" in Lines 292–293 (superscript formatting).

Response: It has been modified as required.

Q7. Reposition the "2.4.6 Transcriptome Sequencing" subsection (Line 295) after Table 3.

Response: The apparent misplacement may reflect software-specific formatting discrepancies. We have reorganized this section to align with journal guidelines.

References:

  1. Wu C, Chen M, Liu S, Yang T, Long L, Guan S, Chen C. Bioactive Flavonoids Icaritin and Icariin Protect against Cerebral Ischemia–Reperfusion-Associated Apoptosis and Extracellular Matrix Accumulation in an Ischemic Stroke Mouse Model. BIOMEDICINES 2021.
  2. Choi J, Ryoo IW, Hong JY, Lee K, Nam HS, Kim WC, Oh S, Kang J, Lee HY, Na S, Heo JH, Lee KO. Clinical impact of estradiol/testosterone ratio in patients with acute ischemic stroke. BMC NEUROL 2021.
  3. Ardelt AA, Anjum N, Rajneesh KF, Kulesza P, Koehler RC. Estradiol augments peri-infarct cerebral vascular density in experimental stroke. EXP NEUROL 2007.
  4. Scheld M, Heymann F, Zhao W, Tohidnezhad M, Clarner T, Beyer C, Zendedel A. Modulatory effect of 17β-estradiol on myeloid cell infiltration into the male rat brain after ischemic stroke. The Journal of Steroid Biochemistry and Molecular Biology 2020.
  5. Wang H, Yang W, Xu L, Han Y, Lin Y, Lu C, Kim K, Zhao Y, Yu X. BV2 Membrane-Coated PEGylated-Liposomes Delivered hFGF21 to Cortical and Hippocampal Microglia for Alzheimer's Disease Therapy. ADV HEALTHC MATER 2024.
  6. Zhu Q, Liu Z, Wang Y, Song E, Song Y. Endoplasmic reticulum stress manipulates autophagic response that antagonizes polybrominated diphenyl ethers quinone induced cytotoxicity in microglial BV2 cells. J HAZARD MATER 2020.
  7. He Z, Yang Q, Li X, Wang Z, Wen S, Dong M, Zhang W, Gong Y, Zhou Z, Liu Q, Dong H. Vanadium Carbide Quantum Dots Exert Efficient Anti-Inflammatory Effects in Lipopolysaccharide-Induced BV2 Microglia and Mice. Small Science 2024.

Round 2

Reviewer 2 Report (New Reviewer)

Comments and Suggestions for Authors
  • I agree with the authors that the TTC-stained sections show improvement in the brain after PAS840 treatment. Although the TTC sections have some technical issues, they are now included as supplementary material. Please add the TTC staining methodology to the manuscript and include the number of animals used in each group.
  • The author used the same animal brains for Evans Blue staining and immunohistochemical analysis. Did the author also use the same brain tissues for RNA analysis? I suggest adding an experimental workflow diagram to make the study design easier to understand in vivo experiments.

Author Response

  • We would like to express our deepest gratitude for your meticulous review and insightful comments on our manuscript. Your valuable assistance serves as a cornerstone in driving the advancement of our research. In response to the two suggestions you raised, we have carefully supplemented and revised the corresponding content with thorough consideration. Please find below our point-by-point responses addressing your specific recommendations. Once again, we reiterate our sincere appreciation for your professional guidance, which has significantly enhanced the quality of this scholarly work.

  • Q1. I agree with the authors that the TTC-stained sections show improvement in the brain after PAS840 treatment. Although the TTC sections have some technical issues, they are now included as supplementary material. Please add the TTC staining methodology to the manuscript and include the number of animals used in each group.
  • Response: We sincerely appreciate your constructive suggestion. In response, we have supplemented the experimental procedures for TTC staining in Section 2.5.6 of the manuscript. Additionally, in Section 3.3.4, we have included explicit guidance directing readers to Figure S1 in the Supplementary Materials for the corresponding staining results.
  • Q2. The author used the same animal brains for Evans Blue staining and immunohistochemical analysis. Did the author also use the same brain tissues for RNA analysis? I suggest adding an experimental workflow diagram to make the study design easier to understand in vivo experiments.
  • Response: We sincerely appreciate your meticulous review in identifying the inconsistencies in our manuscript description. The specimen utilization protocols have been comprehensively revised in Sections 2.5.8 and 2.5.9. Specifically, for Evans Blue staining, we have consistently employed intact cerebral tissue throughout the experimental process. Furthermore, the subsequent histopathological examinations (Hematoxylin-Eosin staining), molecular analyses (PCR), and immunohistochemistry (IHC) were systematically conducted using the same brain tissue specimens. To enhance methodological transparency, we have revised and re-uploaded the complete experimental workflow diagram as the graphical abstract, which will significantly improve readers' comprehension of our technical approach.
  • We extend our deepest gratitude for your invaluable assistance and steadfast support throughout the manuscript evaluation process. Should any additional revisions or clarifications be required to meet scholarly standards, we sincerely hope you will not hesitate to bring them to our attention. Your continued professional guidance would be profoundly instrumental in enhancing the quality of this academic work.

This manuscript is a resubmission of an earlier submission. The following is a list of the peer review reports and author responses from that submission.

Round 1

Reviewer 1 Report

Comments and Suggestions for Authors
  1. Summarize repetitive details, particularly in methods, and focus on crucial experimental steps.
  2. Provide concise descriptions of figures and tables where they are referenced.
  3. Proofread carefully or use professional editing software to refine the text.
  4. The rationale for selecting specific experimental concentrations of PAS840 is not well explained.
  5. The statistical analysis methods (ANOVA, Tukey’s test) are mentioned but lack specific details on how results were validated.
Comments on the Quality of English Language

can be improved. 

Author Response

We deeply appreciate your evaluation of our manuscript, which serves as a significant motivation for our research progress. Through careful review of your insightful comments, we have gained a thorough understanding of the deficiencies in our manuscript. In response, we have carefully revised the content, with particular emphasis on supplementing the preliminary theoretical foundation and experimental basis of this study. Below are our detailed responses to your feedback for your review.

Q1: Summarize repetitive details, particularly in methods, and focus on crucial experimental steps.

A: Thanks for your suggestion, we have made changes to the method section to supplement the method process which is not clearly explained.

Q2: Provide concise descriptions of figures and tables where they are referenced.

A: Thanks for your question. We've gone through all the figures and tables and made clear the unclear descriptions. Moreover, we've added pictures of brain tissue sections from animal brain samples to boost the reliability of the drug's effectiveness.

Q3: Proofread carefully or use professional editing software to refine the text.

A: Thanks for your advice, we used the template provided by the journal Biology for writing, and we also found that we had a mistake in the numbering of some content sections, which we have corrected.

Q4: The rationale for selecting specific experimental concentrations of PAS840 is not well explained.

A: Thank you very much for your suggestion. This was a significant oversight in our writing. We have now clarified in the manuscript how specific experimental concentrations were selected. In Section 2.4.2, we supplemented the rationale for our dosage selection based on CCK-8 assay results, demonstrating that the chosen concentrations effectively promoted cell survival without negative impacts. In Section 2.5.1, we cited our team's previously published in vivo study as a reference to establish the experimental foundation for the dosage selection in our in vivo procedures.

Q5: The statistical analysis methods (ANOVA, Tukey’s test) are mentioned but lack specific details on how results were validated.

A: Thank you for your comments. In section 2.6, we have added the detailed steps of statistical analysis, and also explained that the calculation process of all our data is included in the original data that has been uploaded to the public database, and the analysis report provided by SPSS software is attached. If you need to check our calculation process, you can check our data at the data disclosure at the end of the article. Get the URL for our data.

We sincerely appreciate your critical review and constructive suggestions, which have highlighted significant deficiencies in our manuscript and a major gap in translating our experimental findings to clinical applications. We express our profound gratitude for your invaluable feedback and guidance. We earnestly hope you will review our thoroughly revised manuscript. Should you have any additional comments, we will carefully consider them and make further improvements to our work. Thank you for helping us advance the content of our manuscript.

Reviewer 2 Report

Comments and Suggestions for Authors

This study explores Periplaneta americana extract (PAS840) as a potential treatment for ischemic stroke (IS). PAS840 inhibits inflammation, reduces oxidative stress, enhances neuronal survival, and protects the blood-brain barrier in both cellular and animal models. These findings highlight PAS840’s therapeutic potential for IS. Here are some suggestions below:

·       In the introduction section, the author should provide more details on the causative factors of ischemic stroke models that have been studied in previous literature, as well as specify the target of focus in the current study and explain why it was chosen. Additionally, the author should include more information about PAS840 or related compounds and their effects.

·       The terminology needs to be more consistent throughout the document. For example, there are inconsistencies in the use of "PAS840" and "PAS8440," which should be checked for accuracy. Additionally, the preparation steps for PAS840 could be described in more detail to ensure clarity.

·       The methods for cell viability testing and the OGD/R model should be benefited from more detail. For instance, more details on how OGD/R injury was induced and the specific conditions used for the cell cultures should be useful.

·       Statistical analysis methods need to be clarified. For example, the description of the statistical tests (ANOVA, t-tests) should include more details, such as the software used, significance levels, and how the data was analyzed. This will make the analysis more reproducible.

·       The concentrations of PAS840 (40, 80, 160 μg/mL) are mentioned but not explained in terms of how these doses were determined. It should be more useful to describe whether it reflect a dose-response curve or based on previous research.

·       The author should include more detailed methodology in the methods section. For example, in the biochemical analysis, the author only mentioned performing SOD, NOS, MDA, and ROS assays. Providing more specific details should enhance the reproducibility of the results.

·       The molecular docking methodology could benefit from more details about the parameters used for docking, such as grid size and specific settings. This would make it easier to replicate the analysis and understand the criteria for docking.

·       Figure 13, the whole-brain image with dye lacks sufficient clarity to effectively convey the findings. Adding images of brain sections would enhance the readers' understanding and provide a more detailed visualization of the results.

·       Author does not discuss potential limitations of the study. Addressing limitations, such as the need for further clinical trials or potential side effects, should provide a balanced perspective on the applicability of PAS840 as a therapeutic agent.

·       Overall manuscript is well designed and adding valuable insight in the area of ischemia stroke research.

Author Response

We deeply appreciate your evaluation of our manuscript, which serves as a significant motivation for our research progress. Through careful review of your insightful comments, we have gained a thorough understanding of the deficiencies in our manuscript. In response, we have carefully revised the content, with particular emphasis on supplementing the preliminary theoretical foundation and experimental basis of this study. Below are our detailed responses to your feedback for your review.

Q1: In the introduction section, the author should provide more details on the causative factors of ischemic stroke models that have been studied in previous literature, as well as specify the target of focus in the current study and explain why it was chosen. Additionally, the author should include more information about PAS840 or related compounds and their effects.

A: Thank you for your suggestion. In Lines 61-65 of the manuscript, we have supplemented the primary inducing factors of ischemic stroke (IS) and explained the pathological changes occurring in brain tissue during IS. This addition thereby reinforces that inhibiting oxidative stress and inflammatory responses represents the primary strategy for protecting brain tissue.

Q2: The terminology needs to be more consistent throughout the document. For example, there are inconsistencies in the use of "PAS840" and "PAS8440," which should be checked for accuracy. Additionally, the preparation steps for PAS840 could be described in more detail to ensure clarity.

A: Thank you for bringing this to our attention. This was indeed a significant writing oversight on our part. We have thoroughly proofread the entire manuscript and corrected all instances of "PAS8440," ensuring its complete removal from the text. In Section 2.1, we have provided detailed extraction procedures for PAS840. If our description still lacks precision, we would like to note that although we have filed a patent covering this compound, it cannot be cited at this time due to pending authorization. The patent contains every minute detail of the extraction process, including precise technical parameters and step-by-step protocols.

Q3: The methods for cell viability testing and the OGD/R model should be benefited from more detail. For instance, more details on how OGD/R injury was induced and the specific conditions used for the cell cultures should be useful.

A: Thank you for your reminder. In Sections 2.4.1, 2.4.2, and 2.4.3, we have indeed described in detail the oxygen-glucose deprivation/reoxygenation (OGD/R) model establishment conditions and CCK-8 assay detection methods according to our experimental protocol. These two techniques are well-established experimental methods in the field. Additionally, in Lines 130-107 of the manuscript, we have cited others' OGD/R experiments to supplement any procedural gaps in our own methodology.

Q4: Statistical analysis methods need to be clarified. For example, the description of the statistical tests (ANOVA, t-tests) should include more details, such as the software used, significance levels, and how the data was analyzed. This will make the analysis more reproducible.

A:Thank you for your comments. In section 2.6, we have added the detailed steps of statistical analysis, and also explained that the calculation process of all our data is included in the original data that has been uploaded to the public database, and the analysis report provided by SPSS software is attached. If you need to check our calculation process, you can check our data at the data disclosure at the end of the article. Get the URL for our data.

Q5: The concentrations of PAS840 (40, 80, 160 μg/mL) are mentioned but not explained in terms of how these doses were determined. It should be more useful to describe whether it reflect a dose-response curve or based on previous research.

A: Thank you very much for your suggestion. This was a significant oversight in our writing. We have now clarified in the manuscript how specific experimental concentrations were selected. In Section 2.4.2, we supplemented the rationale for our dosage selection based on CCK-8 assay results, demonstrating that the chosen concentrations effectively promoted cell survival without negative impacts. In Section 2.5.1, we cited our team's previously published in vivo study as a reference to establish the experimental foundation for the dosage selection in our in vivo procedures.

Q6: The author should include more detailed methodology in the methods section. For example, in the biochemical analysis, the author only mentioned performing SOD, NOS, MDA, and ROS assays. Providing more specific details should enhance the reproducibility of the results.

A: Thank you sincerely for your suggestion. We acknowledge that the description of the kit detection procedures in Section 2.4.5 was insufficiently detailed. We have now added supplementary details for all common operational steps. However, since the kit protocols follow standardized and lengthy procedures provided by the manufacturer, we have chosen to maintain originality by including the kit's catalog number (supplied in the Materials section). This allows readers to access the full instructions easily via the company's official website.

Q7: The molecular docking methodology could benefit from more details about the parameters used for docking, such as grid size and specific settings. This would make it easier to replicate the analysis and understand the criteria for docking.

A: Thank you for your suggestion. Given the large number of proteins docked and their varying sizes in our molecular docking experiments, we standardized the description as "grid box volume > 27,000" to streamline the manuscript. However, for readers interested in protein-specific details, we have provided access to our public repository in Lines 232-233 of Section 2.3.7. This repository contains all protein IDs used in the docking experiments, random seed files, and docking box reports (available as both raw files and screenshots).

Q8: Figure 13, the whole-brain image with dye lacks sufficient clarity to effectively convey the findings. Adding images of brain sections would enhance the readers' understanding and provide a more detailed visualization of the results.

A: Your suggestion was absolutely critical to our study. Over the past month, we conducted supplementary hematoxylin-eosin (HE) stained pathological section experiments on all remaining brain tissues from this study. All histological images of these brain sections have been added to Figure 13. Additionally, we supplemented both the experimental procedures and results in Sections 2.5.5 and 3.3.3. This supplementary experiment significantly enhances the credibility of our in vivo findings, and we extend our profound gratitude for your insightful recommendation.

Q9: Author does not discuss potential limitations of the study. Addressing limitations, such as the need for further clinical trials or potential side effects, should provide a balanced perspective on the applicability of PAS840 as a therapeutic agent.

A: This represents a significant deficiency in our manuscript writing and a major flaw in translating our experimental findings to clinical application. We have addressed this by supplementing Lines 781-787 in the discussion section. We express our profound gratitude for your insightful feedback and suggestions.

We sincerely hope you will review our thoroughly revised manuscript. Should you have any additional comments, we will carefully consider them and make further improvements to our work. Thank you for helping us advance the content of our manuscript.